# Spatial variability of the modern radiocarbon reservoir effect in the high-altitude lake Laguna del Peinado (Southern Puna Plateau, Argentina)

Paula A. Vignoni[1, 2], Francisco E. Córdoba[3, 4], Rik Tjallingii[2], Carla Santamans[3,4], Liliana C. Lupo[3, 5], Achim Brauer[1, 2]

[1]Institut für Geowissenschaften, Universität Potsdam, Potsdam, 14476, Germany.
[2]Klimadynamik und Landschaftsentwicklung, Deutsches GeoForschungsZentrum GFZ, Potsdam, 14473, Germany.
[3]Instituto de Ecorregiones Andinas INECOA, CONICET – Universidad Nacional de Jujuy, San Salvador de Jujuy, 4600, Argentina.
[4]Instituto de Geología y Minería, Universidad Nacional de Jujuy, San Salvador de Jujuy, 4600, Argentina.
[5]Facultad de Ciencias Agrarias, Universidad Nacional de Jujuy, San Salvador de Jujuy, 4600, Argentina.

*Correspondence to*: Paula A. Vignoni (pvignoni@gfz-potsdam.de, vignoni.paula@gmail.com)

**Abstract.** The high-altitude lakes of the Altiplano-Puna Plateau in the Central Andes commonly have large radiocarbon reservoir effects. This, combined with the general scarcity of terrestrial organic matter makes obtaining a reliable and accurate chronological model based on radiocarbon ages a challenge. As a result, age-depth models based on radiocarbon dating are often constructed by correcting for the modern reservoir effect, however, commonly without consideration of spatial and possible temporal variations of reservoir ages within the lake and across the basin. In order to get a better constraint on the spatial variability of the radiocarbon reservoir effects, we analyse [14]C ages of modern terrestrial and aquatic plants from the El Peinado basin in the Southern Puna Plateau, which hosts the Laguna del Peinado lake fed by hydrothermal springs. The oldest [14]C ages of modern samples (>18,000 and >26,000 BP) were found in hot springs discharging into the lake likely resulting from the input of [14]C-depleted carbon from old groundwater and [14]C-free magmatic $CO_2$. In the littoral and central part of Laguna del Peinado, modern samples [14]C ages were several thousand years younger (>13,000 and >12,000 BP) compared to the inflowing waters as a result of $CO_2$ exchange with the atmosphere. Altogether, our findings reveal a spatial variability of up to 14,000 [14]C years of the modern reservoir effect between the hot springs and the northern part of the Peinado lake basin. Temporal changes of reservoir effects in sediment records are more difficult to quantify but [14]C ages from a short core from Laguna del Peinado may suggest temporal reservoir age variations of a few thousand years. This study has implications for accurate [14]C-based chronologies for paleoclimate studies in the Altiplano-Puna Plateau and similar settings. Our results highlight the need to consider spatial and likely also temporal variations in the reservoir effects when constructing age-depth models.

 **1 Introduction**

The Altiplano-Puna Plateau is the second highest and largest plateau in the world after Tibet (Allmendinger et al., 1997). It extends along the Central Andes from southern Peru to northern Argentina and Chile at an altitude above 3,000 m a.s.l. and is characterised by endorheic basins that host numerous saline lakes, playa-lakes, and salars. These systems are highly sensitive to small fluctuations in the water balance making them promising sensors for studying recent and past environmental and hydrological changes (e.g. Grosjean et al., 1997; Valero-Garcés et al., 2000; McGlue et al., 2013; Santamans et al., 2021) associated with the dynamics of the South American Monsoon System and the more northerly influence of the Southern Hemisphere Pacific Westerlies (Vuille and Ammann, 1997; Garreaud, 2009; Vuille et al., 2012).

Our understanding of the regional and temporal hydroclimatic dynamics in the Altiplano-Puna Plateau is hampered by the difficulty in obtaining accurate chronologies from lacustrine sediments due to the scarcity of terrestrial organic matter and the anomalously old apparent $^{14}$C age of waters and hence aquatic samples, known as 'reservoir effect' (Grosjean et al., 1995, 1997, 2001; Geyh et al., 1998; Valero-Garcés et al., 2000; Yu et al., 2007). In particular, in the southern sector of the Andean Plateau (i.e. the Puna region; Fig. 1), no other paleoclimatic archives (e.g. ice-glaciers) exist that can be studied and used to develop high-resolution paleoclimatic and paleoenvironmental reconstructions like those located further north in the Altiplano (e.g. Thompson et al., 1995, 1998, 2000; Zech et al., 2008). Therefore, obtaining reliable chronological models using lake sediments from this region is critical and requires an understanding of the $^{14}$C reservoir effect variability in each particular lake system. Reservoir effects depend on different causes including $CO_2$ exchange rates between the water and the atmosphere, the internal system mixing dynamics, and the input of $^{14}$C-free ('dead') or $^{14}$C-depleted carbon either derived from dissolved carbonates, volcanic $CO_2$ or the inflow of old groundwater (Macdonald et al., 1991; Ascough et al., 2010; Keaveney and Reimer, 2012; Jull et al., 2013; Lockot et al., 2015). Since the application of other dating methods is also strongly limited or not possible (like $^{210}$Pb and $^{137}$Cs due to the low concentration of these isotopes at these latitudes and altitudes; Argollo et al., 1994; Cisternas and Araneda, 2001), the construction of chronological models based on corrections for the modern reservoir effect is a common practice (e.g. Grosjean et al., 1995, 1997, 2001; Geyh et al., 1998, 1999; Moreno et al., 2007; Giralt et al., 2008; Jara et al., 2019). Sometimes, even assumptions on temporal variations of the reservoir effect are included in the construction of age-depth models (e.g. Grosjean et al., 2001; Moreno et al., 2007). In contrast, spatial variations within a lake system are less considered although these can be equally crucial for reliable age-depth models.

Therefore, the aim of this study is to present and discuss new radiocarbon data from various types of modern samples from a range of different locations within the Laguna del Peinado lake system and catchment. This high-altitude lake in the Southern Puna Plateau is fed by hydrothermal springs and previous studies assumed a modern reservoir effect >12,000 years for a lake sediment record (Valero-Garcés et al., 1999, 2000). We analysed the $^{14}$C ages of live/modern terrestrial and aquatic plants within the El Peinado basin to investigate whether significant variability exists and if it follows a spatial trend, and what are

the likely sources of the reservoir effect. Furthermore, we report down-core $^{14}$C ages from a lake core and compare them with the previously published dates to estimate possible temporal variability of the reservoir effect in the sedimentary record of this lake.

## 2 Study area

Laguna del Peinado is a shallow lake located at 3,760 m a.s.l. in the Southern Puna Plateau in the Central Volcanic Zone of the Andes of Argentina (26° 30' 16.87" S, 68° 5' 49.95" W; Fig. 1). The lake is located in a topographically closed basin along the Peinado lineament, a NNE-SSW dextral transcurrent fault system that runs along the Antofalla salar axis and presumably continues under the Peinado composite volcano (Seggiaro et al., 2006; Grosse et al., 2022). Peinado is a young potentially active volcano (last activity dated to 36.8 ± 3.8 ka; 5,890 m a.s.l.) located ~5 km south of Laguna del Peinado (Grosse et al.,

2022). It consists of a very steep cone surrounded by a ring of lavas originating from several lateral vents, loose scoria and subordinate pyroclastic flows of dominantly mafic compositions (basaltic andesites and andesites; Grosse et al., 2014, 2017, 2022). The Peinado volcanic field also hosts 17 mafic monogenetic centres, with seven of them located between the Antofalla salar and the Peinado stratovolcano and of Pleistocene ages (0.6 ± 0.1 Ma to 0.15 ± 0.02 Ma; Grosse et al., 2020). Lavas from one of these monogenetic centres (0.38 ± 0.02 Ma; Grosse et al., 2020) limit the southern part of the Laguna del Peinado

system (Fig. 1). To the west of the lake, the Pliocene rhyolitic Laguna Amarga ignimbrites (3.7-4 Ma, >70 km$^3$) originated from the Laguna Amarga caldera crop out (Kay et al., 2010 and references therein; Fig. 1). Four lacustrine terraces have been described, the uppermost consisting of volcanic sandstones and conglomerates cemented with calcite, and the lower ones composed by intraclastic and biomicritic limestones (Valero-Garcés et al., 2001).

The main water body in the El Peinado basin is Laguna del Peinado with a maximum length in the N-S direction of 3.4 km, a

width of 1.2 km, and a maximum water depth of ~4 m (Fig. 1). The lake is fed by hydrothermal springs located on the southern and western shores. The southern part of the lake is characterised by an extensive shallow wetland area where hot spring seeps and a pool discharge through a stream into the lake, while on the western shore several smaller hot spring seeps occur. To the north of Laguna del Peinado, the smaller and shallower Laguna Turquesa is located which currently is disconnected due to the low water level (Fig. 1). Both lakes were connected until ca. 2005 according to satellite images (Villafañe et al., 2021).

Carbonate precipitation takes place within both lakes and the hydrothermal springs environments as a result of $CO_2$ degassing, evaporation, and biological processes with deposits comprising a wide variety of facies including microbialites (travertines, microbial mounds, microbial mats, and others) and fine-grained mineral precipitates (Valero-Garcés et al., 2001; Farías et al., 2020; Della Vedova et al., 2022; Vignoni et al., 2022).

The location of the El Peinado basin in the southern portion of the Puna Plateau covers the climatic transition zone between

the South American Monsoon System (SAMS) and the more northerly influence of the Southern Hemisphere Pacific

Westerlies (SHPW; Fig. 1). Due to the meridional extension and prominent orography, the Central Andes act as a topographic barrier to the moisture-bearing easterly winds resulting in a steep E-W rainfall gradient with high precipitation on the eastern flanks and increasing aridity westwards into the Puna Plateau (Strecker et al., 2007; Garreaud, 2009; Castino et al., 2017). In the study region, mean annual precipitation values are <120 mm yr$^{-1}$, whereas evaporation has been estimated to be >1,500 mm yr$^{-1}$ in Laguna del Negro Francisco located ~150 km southeast of Laguna del Peinado (Grosjean et al., 1997; Strecker et al., 2007). The scarce precipitation events that reach these dry high elevation zones are associated to seasonal changes in the position and intensity of the two dominant atmospheric circulation systems (SAMS vs. SHPW; Fig. 1). Approximately 80% of the annual precipitation occurs during summer when the strengthening of the SAMS and the heating effect over the Puna Plateau are responsible for convective rainfall events (Garreaud, 2009; Vuille et al., 2012). During winter, precipitation is associated with northward incursions of the SHPW and snowfall events result from Pacific cold fronts often combined with blocking episodes in the South Pacific and polar air isolated cells that migrate north interacting with warmer tropical air masses (Vuille and Ammann, 1997).

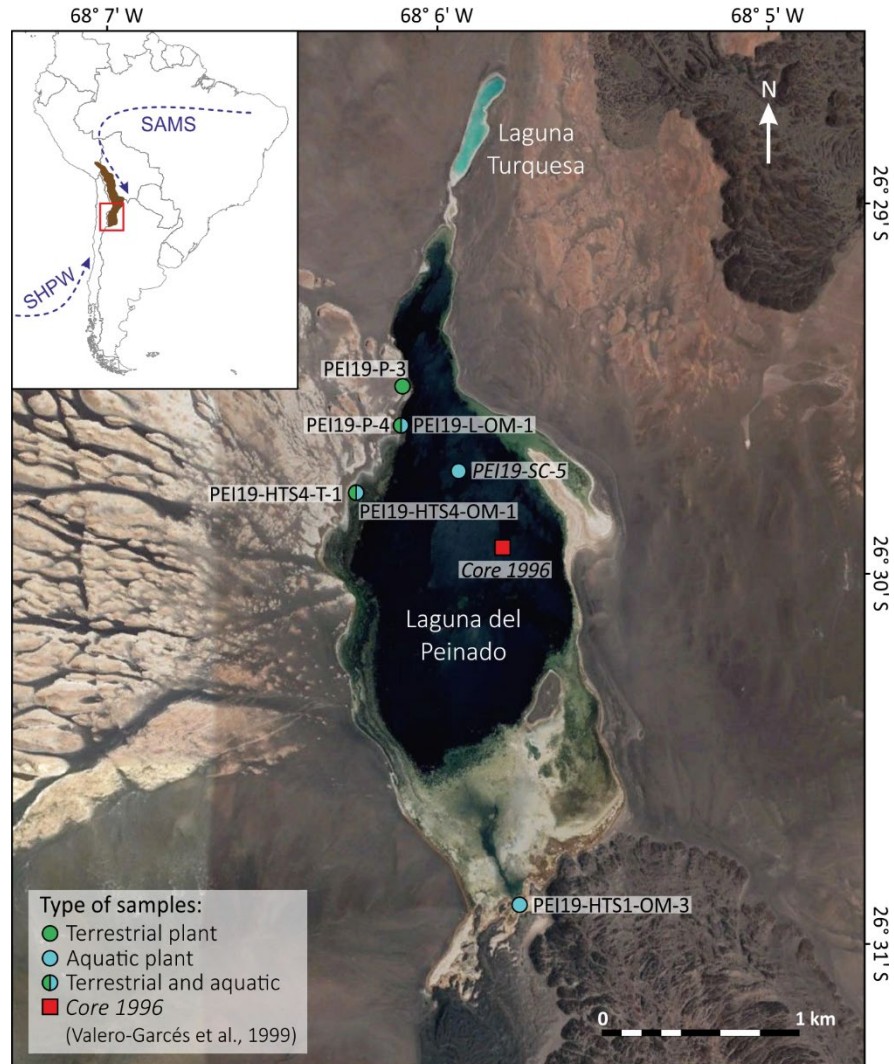

**Figure 1: Location and type of samples collected in the El Peinado basin during 2019 (© Google Earth 2020, Maxar Technologies, CNES/Airbus). Sediment core samples are indicated in italics. Left top corner: map of South America with the Altiplano-Puna Plateau highlighted in brown and the climatic moisture sources (SAMS: South American Monsoon System, SHPW: Southern Hemisphere Pacific Westerlies). The red square marks the approximate location of the El Peinado basin in the Puna Plateau of NW Argentina.**

## 3 Materials and methods

### 3.1 Fieldwork and sampling

During fieldwork in January and November 2019, sediment and organic matter samples including living/modern terrestrial and aquatic plants were collected for radiocarbon analysis from different parts of the lake basin (Fig. 1). Short sediment cores

(<1 m) were retrieved from Laguna del Peinado in two profiles at different water depths using an Uwitec coring device. The cores were split into two halves, photographed and described at the GFZ Potsdam (Germany). Four bulk samples were sieved for organic macro remains used for radiocarbon dating from a short core in the central part of the basin at 3.2 m water depth (PEI19-SC-5, Fig. 1). In total, we collected 10 samples: seven modern samples (three terrestrial plants, two microbial mats from the hot springs, aquatic macrophytes from the lake littoral zone, and aquatic macrophytes from the sediment core-top at 0-2 cm depth) and three plant macrofossil samples (from the sediment core at 22-23, 48-49, and 71-72 cm depth). All samples were washed with demineralised water and dried in the oven at 60 °C. Plant materials are reported in Table 1 and modern samples are shown in Fig. 2.

## 3.2 Radiocarbon and carbon isotopes analysis

Samples preparation, chemical pre-treatments, and accelerator mass spectrometry (AMS) $^{14}$C measurements were carried out in the Poznan Radiocarbon Laboratory (Poland). A full description of the procedures can be accessed at: https://radiocarbon.pl/en/sample-preparation/. After mechanical removal of macroscopic contamination under binoculars, the samples underwent a sequential acid-base-acid (ABA) treatment following the protocols established for each material (UW protocol for the wood sample PEI19-P-3 and UV protocol for all other plant samples). Samples were first treated with 1 M HCl at 80°C for 20 min or longer if needed until gas bubbles emanations finished (UV, UW), followed by 0.1M NaOH treatment at room temperature for fragile plant remains (UV) and 80°C for wood (UW); and then 0.25M HCl at 80°C for 1 hr. After each treatment, samples were rinsed with deionised water (Millipore) to pH=7. The NaOH treatment step is repeated a few times until no more colouring of the solution caused by humic acids is observed. For the wood sample PEI19-P-3 (UW), an additional treatment with 5% NaClO$_2$ at room temperature was applied for 30 min. The resulting $^{14}$C ages are listed in Table 1 with one standard deviation (σ). Samples with percentage of modern carbon (pMC) and radiocarbon ages were converted to fraction modern $^{14}$C (F$^{14}$C) values (Stuiver and Polach, 1977; Reimer et al., 2004; Stenström et al., 2011) using the R package 'rintcal' (Blaauw, 2003). Two post-bomb dates from terrestrial plant samples collected at the shore without contact to lake water were calibrated with CALIBomb (Reimer et al., 2004; Reimer and Reimer, 2023) using the Southern Hemisphere Zone 1-2 calibration data set (Hua et al., 2022).

We conducted a simple end-member mixing model to calculate the approximate proportion of dead ($^{14}$C-free) versus modern (atmospheric) carbon in each sample following Pasquier-Cardin et al. (1999) as:

$$Dead\ carbon\ (\%) = [1 - (\text{F}^{14}\text{C in sample}/\text{F}^{14}\text{C in reference plant})] \times 100 \quad (1)$$

We considered sample PEI19-P-4 as the reference plant that best represents local atmospheric F$^{14}$C (Table 1) at the time of sampling (2019) compared to the average value for Southern Hemisphere Zone 1-2 (1.019; Hua et al., 2022). We assumed that the $^{14}$C content in this sample was in equilibrium with the local atmospheric carbon.

Additionally, $\delta^{13}C_{carb}$ was analysed in four samples from the carbonatic matrix sediments at 0-2, 24-26, 46-48, and 71-72 cm depth from the core where the plant macrofossils have been taken, and in one sample from the microbial mats in the southern hot spring. Samples were frozen for 24 to 48 hours, freeze-dried for 72 hours, and ground to powder. Carbon isotopes analysis of carbonate powders ($\delta^{13}C_{carb}$) were carried out on an automated carbonate extraction device (KIEL IV) coupled to a Finnigan MAT 253 IRMS (Thermo Fisher Scientific) at the GFZ Potsdam. In brief, acid digestion of carbonates with phosphoric acid takes place in the KIEL IV to produce $CO_2$ that is ultimately analysed for $\delta^{13}C_{carb}$ in the coupled MAT 253 IRMS. Results are expressed in the conventional $\delta$-notation in per mille (‰) relative to VPDB (Vienna Pee Dee Belemnite; Table 1). Repeated measurements of the reference material NBS 19 ensured an analytical precision better than $\pm$ 0.07‰ ($\sigma$).

## 4 Results

Radiocarbon dating results of modern terrestrial and aquatic plants showed a wide variety of ages in the lake and catchment area (Table 1). Only two samples of terrestrial plants located approximately 15 and 5 m away from the lake shore gave modern ages, PEI19-P-3 (*Adesmia sp.,* 112.39 $\pm$ 0.31 pMC) dated to 1994-1996 cal CE and PEI19-P-4 (Poaceae, 101.61 $\pm$ 0.34 pMC) to 2018-2019 cal CE (Fig. 1, 2a, and 2b). Another terrestrial plant (Poaceae) collected near a hot spring (~15 cm) on the western shore (Fig. 1 and 2d) resulted in an age of 1,580 $\pm$ 30 BP. In contrast, a microbial mat sample of this hot spring dated to 18,510 $\pm$ 90 BP (Fig. 1 and 2c). The oldest measured age was found in microbial mats from the southern shore hot spring pool (26,500 $\pm$ 1300 BP; Fig. 1 and 2e). Living aquatic macrophytes (Potamogetonaceae) on the northwestern lake shore (~20 cm water depth) and from the surface sediment taken from the core located ~370 m away (3.2 m water depth) showed similar ages of 13,840 $\pm$ 70 BP and 12,360 $\pm$ 60 BP, respectively (Fig. 1, 2f and 2g). The three down-core samples from aquatic macrophytes (Potamogetonaceae) are in chronostratigraphic order and revealed ages of 16,180 $\pm$ 80 BP (22 to 23 cm), 16,720 $\pm$ 90 BP (48 to 49 cm), and 17,680 $\pm$ 90 BP for the basal layer (71 to 72 cm; Fig. 3).

A simple end-member mixing model revealed highest proportion of dead carbon in modern microbial mats from the southern and western hot springs (96.4% and 90.2%, respectively), while values for the lake modern aquatic macrophytes ranged between ~78 and 82% (Table 1, Fig. 2).

Table 1: $^{14}C$ ages from El Peinado basin. $F^{14}C$ values were calculated with the package 'rintcal' (Blaauw, 2003). The proportion of dead ($^{14}C$-free) carbon was calculated with reference to sample PEI19-P-4, considered representative the local atmospheric $F^{14}C$. As a reference, for the year 2019 when these samples were collected, the mean value of atmospheric $F^{14}C$ for the Southern Hemisphere Zone 1-2 is 1.019 (January to May; Hua et al., 2022). The $\delta^{13}C$ values in italic correspond to samples at 24 to 26 cm and 46 to 48 cm, and differ at the sampling depths for $^{14}C$. Question marks (?) denote samples where water influence, water mixing, and plants genus and/or species could not be determined with certainty.

| Sample | Water influence and type | Material | Lab ID | Lab reported $^{14}C$ age ± σ (BP) | $F^{14}C$ ± σ | Proportion of dead carbon (%) | $\delta^{13}C_{carb}$ (‰ VPDB) |
|---|---|---|---|---|---|---|---|
| PEI19-P-3 | No water influence, ~2 m above lake level | Terrestrial plant (*Adesmia horrida?*, 'Añagua') | Poz-122590 | 112.39 ± 0.31 pMC | 1.1239 ± 0.0031 | - | - |
| PEI19-P-4 | No water influence, ~1 m above lake level | Terrestrial plant (Poaceae, *Festuca ortophylla?*) | Poz-123623 | 101.61 ± 0.34 pMC | 1.0161 ± 0.0034 | 0 | - |
| PEI19-L-OM-1 | Lake water (littoral, ~20 cm below water level) | Aquatic macrophyte (Potamogetonaceae, *Potamogeton sp.* or *Zannichellia sp.?*) | Poz-123624 | 13,840 ± 70 | 0.1786 ± 0.0016 | 82.4 | - |
| PEI19-HTS4-T-1 | Hot spring 4? (~15 cm away from the shore) | Terrestrial plant (Poaceae, *Festuca ortophylla?*) | Poz-123625 | 1,580 ± 30 | 0.8215 ± 0.0031 | 19.2 | - |
| PEI19-HTS4-OM-1 | Hot spring 4 (shallow, mix with lake water?) | Microbial mats in hot spring | Poz-122473 | 18,510 ± 90 | 0.0998 ± 0.0011 | 90.2 | - |
| PEI19-HTS1-OM-3 | Hot spring 1 southern shore (pool bottom) | Microbial mats in hot spring | Poz-123626 | 26,500 ± 1,300* | 0.0369 ± 0.0055 | 96.4 | 7.09 |
| PEI19-SC-5_0 to 2 cm | | | Poz-132432 | 12,360 ± 60 | 0.2147 ± 0.0016 | 78.9 | 8 |
| PEI19-SC-5_22 to 23 cm | Lake water (short core, 3.2 m water depth) | Aquatic macrophyte (Potamogetonaceae, *Potamogeton sp.* or *Zannichellia sp.?*) | Poz-132519 | 16,180 ± 80 | - | - | *8.6* |
| PEI19-SC-5_48 to 49 cm | | | Poz-132520 | 16,720 ± 90 | - | - | *9.2* |
| PEI19-SC-5_71 to 72 cm | | | Poz-132433 | 17,680 ± 90 | - | - | 8.9 |
| Core 1996 (Valero-Garcés et al., 1999) | Lake water (core) | Macrophyte (0 to 1 cm) | WHOI 17536 | 12,750 ± 90 | - | - | - |

*The high σ is due to the small sample size (0.05 mgC).

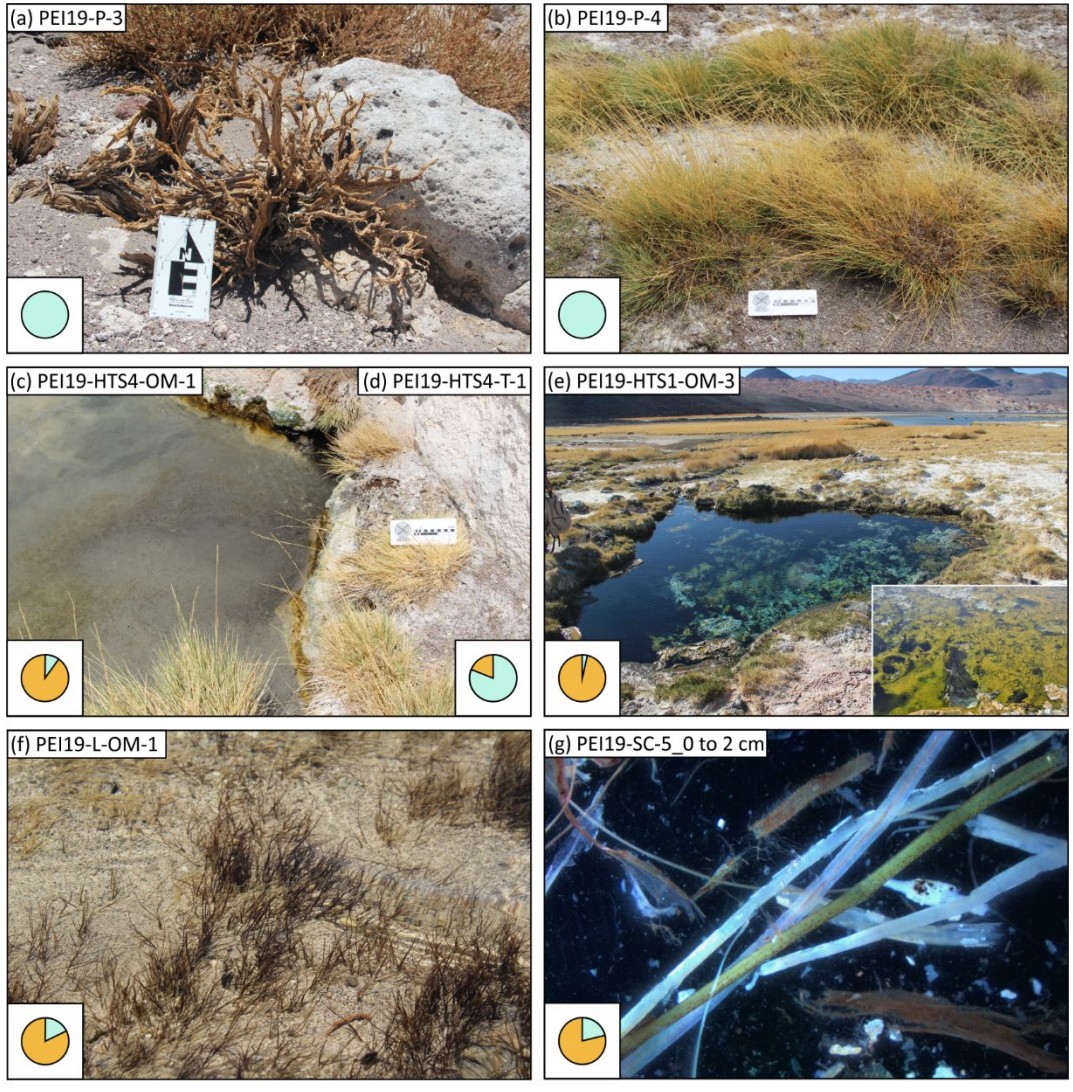

**Figure 2: Modern samples: (a) and (b) terrestrial, (c) aquatic and (d) terrestrial by the western shore hydrothermal spring, (e) aquatic from the southern shore hydrothermal spring, (f) lake littoral, (g) aquatic from the top of the lake short core. The pie charts in the bottom corners show the estimated proportion of modern and dead carbon for each sample (Table 1).**

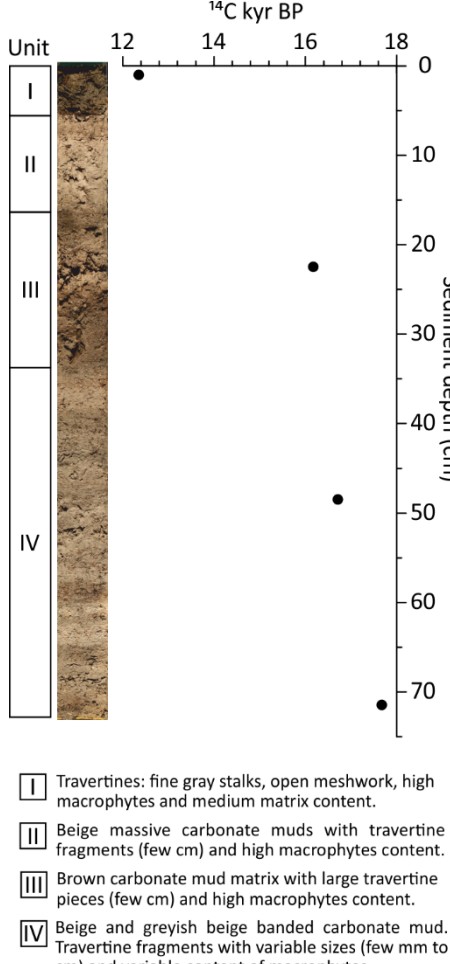

**Figure 3: Short core PEI19-SC-5: down-core non-corrected ¹⁴C ages obtained from aquatic macrophytes (Potamogetonaceae) remains (see Table 1). The top sample corresponds to live aquatic macrophytes (Potamogetonaceae).**

## 5 Discussion

In this section, we discuss the variability observed in the ¹⁴C ages of modern plants in the El Peinado basin and the possible sources of ¹⁴C-free or ¹⁴C-depleted carbon causing reservoir effects that may also be of relevance for other basins in the
185 Altiplano-Puna Plateau as well as basins in similar settings.

### 5.1 Spatial variability of the ¹⁴C reservoir effect in the El Peinado basin

Present-day terrestrial plants are commonly expected to provide modern radiocarbon ages, while aquatic plants potentially take up old carbon. The only two terrestrial plant samples in our study that yielded recent ages and were free of any reservoir effects were found 15 and 5 m away from the lake shore line (Fig. 1 and 4). PEI19-P-3, a woody plant of the genus *Adesmia,*

was most likely dead at the time of sampling as it had no sprouts (Fig. 2a, the front plant was sampled) explaining the high

$F^{14}C$ value of $1.1239 \pm 0.0031$ (Table 1) which corresponds to the atmospheric $F^{14}C$ of year 1994-1996 CE (Hua et al., 2022).

PEI19-P-4, a Poaceae possibly *Festuca ortophylla*, dated to 2018-2019 cal CE in agreement with the sampling year (Table 1,

Fig. 2b). Another Poaceae sample (PEI19-HTS4-T-1) growing in the vicinity of a hydrothermal spring (~15 cm) revealed an

age of $1,580 \pm 30$ BP indicating incorporation of $^{14}C$-depleted carbon (~20%; Table 1, Fig. 2d). The ages of all modern aquatic

samples varied substantially between $12,360 \pm 60$ BP and $26,500 \pm 1300$ BP, showing distinct reservoir effects depending on

the location within the basin. The oldest age was found for the modern microbial mats from the southern shore hydrothermal

pool, whereas microbial mats from a hot spring at the western shore displayed an approximately 8,000 $^{14}C$ years younger age

(Table 1, Fig. 1 and 4). Ages from aquatic plants from a near-shore site and surface sediment from the central part of the lake

reveal significantly younger ages than those from microbial mats in both hot springs (Table 1, Fig. 1 and 4). From those two,

the shallow water sample is ca. 1,500 years older ($13,840 \pm 70$ BP) than the sample from the central part of the lake ($12,360 \pm$

60 BP). The general trend of decreasing ages from the inflowing hydrothermal waters and littoral positions towards the deeper

lake likely reflects longer residence time of water in the lake and prolonged exchange of the dissolved inorganic carbon (DIC)

with the atmospheric $CO_2$ compared to the hydrothermal inflows, resulting in higher $^{14}C$ concentrations and consequently

lower reservoir effects (Table 1, Fig. 1 and 4). In contrast, the higher reservoir effect recorded in the plants from the hot springs

and the littoral zone of the lake reflects poorly equilibrated DIC. The difference of ca. 8,000 $^{14}C$ years between both hot springs

could result either from some influence (i.e. mixing) of lake water with higher $^{14}C$ concentrations in the western shore hot

spring, or from the existence of separate hydrothermal systems bounded by the Peinado lineament with distinct $^{14}C$ content in

the DIC (Table 1, Fig. 1 and 4). A similar pattern of spatial variability has been observed in lacustrine systems in the Tibetan

Plateau, with high reservoir effect in tributaries and spring waters and lower reservoir effect in the central regions of lakes with

differences of up to 19,000 $^{14}C$ years between different locations within individual systems (Mischke et al., 2013).

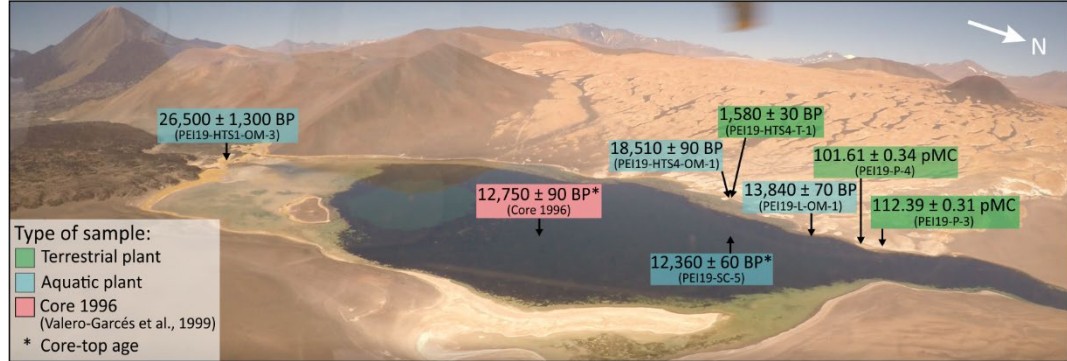

**Figure 4: Aerial view of Laguna del Peinado from the northeast and all radiocarbon dates obtained from modern surface samples. For a top view, please refer to Figure 1.**

## 5.2 $^{14}$C reservoir effect sources in the El Peinado basin

### 5.2.1 Catchment geology

The dissolution of carbonate-rich sediments or rocks in the catchment area is usually considered a main source of $^{14}$C-free carbon influx into a lake (Macdonald et al., 1991; Ascough et al., 2010). However, the dissolution of catchment carbonates can only be a minor source of $^{14}$C-free carbon into Laguna del Peinado because the lithology of the basin is dominated by volcanic rocks (Fig. 1; Seggiaro et al., 2006; Grosse et al., 2020, 2022) and extreme arid conditions prevail. Carbonatic outcrops are scarce and limited to few ancient lake terraces consisting of conglomerates and sandstones with volcanic clasts
cemented/coated with calcite, intraclastic and biomicritic limestones, and microbialites (Valero-Garcés et al., 2001; Villafañe et al., 2021). Furthermore, calcium available for carbonate formation in this lacustrine system is interpreted to derive from the alteration of the volcanic bedrock by fluids at high temperatures as described in other lake systems in the Altiplano-Puna Plateau (e.g. Laguna Pastos Grandes; Muller et al., 2020).

### 5.2.2 Groundwater effects

Another common source of $^{14}$C-depleted carbon in lakes can be the contribution of old groundwater (Macdonald et al., 1991; Riggs, 1984; Godfrey et al., 2021). In the Altiplano-Puna Plateau, the contribution of old groundwater to lacustrine systems might be substantial given the present-day negative water balance regime (Grosjean et al., 1995). Furthermore, water tables can be very deep and develop groundwater flow paths with long transit times that often cross topographic boundaries prior to emerging at the basin bottoms (Moran et al., 2021). The old $^{14}$C ages of modern aquatic organisms in the hot springs (Table 1,
Fig. 4), can be partially explained by an old origin and high residence times of water in the hydrothermal groundwater system. This is supported by geochemical studies suggesting a meteoric origin of water in the hydrothermal reservoir (Vignoni et al., 2022), which must have occurred in the past given the extreme low present-day precipitation. Groundwater in this region is thought to have formed during wet periods at the end of the last glaciation and early Holocene like the widespread Central Andean Pluvial Event (17.5–14.2 ka and 13.8–9.7 ka; Latorre et al., 2006; Placzek et al., 2009; Gayo et al., 2012), which
overlap with the ages obtained for modern samples from the northwestern shore and the top of the lake core. Water from these wet periods, however, would be too young to explain the two older dates of $18,510 \pm 90$ BP and $26,500 \pm 1,300$ BP obtained from organisms directly in the hot springs (Table 1, Fig. 4). Therefore, additional sources of $^{14}$C-free or $^{14}$C-depleted carbon must be involved (see Section 5.2.3).

The influence of old groundwater in the lake is consistent with $^{3}$H analysis in wetland systems in the Southern Puna Plateau
proving that these environments are mainly sustained by old waters that can be centuries to several millenia old, with only minor contribution of modern water (<60 years old) not exceeding 10% (Moran et al., 2019, 2021; Frau et al., 2021). For example, the zero $^{3}$H activity in the lake waters of the Carachi Pampa basin (~50 km east of the El Peinado basin) indicates

that it is almost entirely sustained by old groundwater (Frau et al., 2021). Also, geochemical ($^3$H, $\delta^{18}$O, $\delta^2$H) and hydrophysical studies in springs and groundwater feeding the Salar de Atacama basin, located ~325 km north of the study area, revealed a large regional groundwater system integrated over timescales of 100-10,000 years or longer (Moran et al., 2019). Although $^3$H data of waters is lacking for the El Peinado basin, this system most likely is almost entirely supported by old groundwater as reported also from other sites in the region (e.g. Frau et al., 2021; Moran et al., 2019, 2021; Godfrey et al., 2021).

### 5.2.3 $^{14}$C-free volcanic $CO_2$

Another potential source for the large reservoir effects in El Peinado basin, in particular for those dates exceeding the ages of the Central Andean Pluvial Events, is a contribution of $^{14}$C-free magma derived $CO_2$ (Macdonald et al., 1991 and references therein; Pasquier-Cardin et al., 1999). In the young volcanic El Peinado basin (Grosse et al., 2020, 2022), magma degassing is a likely source of $CO_2$ into the hydrothermal system as assumed for the Cerro Blanco geothermal system located 40 km to the ESE (Chiodi et al., 2019). Geophysical studies also suggested the existence of magma reservoirs in the crust beneath the El Peinado area (Bianchi et al., 2013; Ward et al., 2017). An influence of volcanic $CO_2$ is further supported by permanent bubbling in the southern shore hot spring pool as well as in the seepage areas on the west coast. Furthermore, strong $^{13}$C-enrichment in the lake carbonates up to +13‰ VPDB has been explained by degassing of volcanic $CO_2$ (Table 1; Valero-Garcés et al., 1999). Considering the scenario of different hydrothermal systems on the south and west lake coasts, a higher contribution of magmatic $CO_2$ in the southern hydrothermal reservoir compared to those on the west would explain the oldest age of 26,500 ± 1,300 BP recorded in the basin. For example, $^{14}$C ages of the DIC exceeding 20,000 years due to dilution by geogenic DIC sources have been reported also from the Loa and Calama basins located north of the Salar de Atacama (~450 km NNW of the El Peinado basin; Godfrey et al., 2021). Moreover, the aquatic plant with the oldest $^{14}$C age has a proportion of modern carbon (~ 4%; Table 1, Fig. 2d) supporting that the reservoir ages result from dilution with $^{14}$C-free volcanic $CO_2$.

Degassing of magmatic $CO_2$ might have also caused the age of 1,580 ± 30 BP for a modern terrestrial plant from the western shore (Table 1, Fig. 1 and 2c). It has been reported that diffuse emanations of magmatic $CO_2$ through soils lead to a substantial $^{14}$C depletion in terrestrial plants when $^{14}$C-free $CO_2$ is assimilated during photosynthesis (Pasquier-Cardin et al., 1999). This might explain the old age of the terrestrial plant sample since it grew at a distance of only ~15 cm from the local hot spring. Potential uptake of soil DIC through the roots might additionally contribute but only to a very minor degree since it usually represents less than 1% of the total $CO_2$ fixed by plants (Loczy et al., 1983; Brix, 1990; Enoch and Olesen, 1993; Ford et al., 2007). The other two dated terrestrial plants were not affected by volcanic $CO_2$ contamination probably because they grew about 5 m and 15 m further away from the lake shore and from potential sources of volcanic $CO_2$ as no hot springs were identified in that area (Table 1; Fig. 2a, 2b, and 4).

### 5.3 $^{14}$C reservoir effect in surface sediments

The core-top age obtained from aquatic macrophytes (Potamogetonaceae) from core PEI19-SC-5 (12,360 ± 60 BP) is ~400 $^{14}$C years younger compared to that of a core taken in 1996 (Valero-Garcés et al., 1999, 2000; Table 1, Fig. 4). A likely explanation for a larger reservoir effect in the previous core is its location closer to the southern hydrothermal spring. This would be in agreement with our observations in the modern environment revealing decreasing reservoir effects with increasing distance to the hydrothermal springs due to extended DIC equilibration with atmospheric $CO_2$.

However, we cannot fully exclude a decrease in the reservoir effects in the last 23 years since the core from Valero-Garcés et al. (1999) was recovered probably related to a lake level lowering of at least 0.6 m and the associated disconnection between Laguna del Peinado and Laguna Turquesa (Villafañe et al., 2021). This has been assumed for other lakes in the Altiplano-Puna Plateau (e.g. Laguna Lejía, Laguna Miscanti, Lago Chungará; Geyh et al., 1998, 1999; Grosjean et al., 2001; Moreno et al., 2007; Giralt et al., 2008). In contrast, studies from other lake environments in NW China report an opposite mechanism, i.e. increasing reservoir effect related to lake level lowering (Zhou et al., 2020). Due to the lack of systematic studies, the influence of short-term lake level fluctuations on modern reservoir effects remains elusive. Other potential causes for temporal reservoir effect changes might be fluctuations in geothermal activity (Ascough et al., 2010) or changes in the lake primary productivity (Zhou et al., 2020), though we have no evidence to investigate these in more depth.

### 5.4 Down-core $^{14}$C ages

The four down-core $^{14}$C ages from aquatic macrophytes (Potamogetonaceae) in our core PEI19-SC-5 are in chronological order (Table 1, Fig. 3). The three radiocarbon ages between ca. 22 and 72 cm core depth suggest that these 50 cm of sediment comprise ca. 1,500 years (Fig. 3). The age of 12,360 ± 60 BP from the core surface suggests that the uppermost 22 cm of the sediment record cover a period of ca. 4,000 years. Assuming a largely constant reservoir effect, either the sedimentation rate must have decreased or the sediment record includes a major hiatus. We do not observe lithological indications in the sediment core neither for a substantial sedimentation rate change nor for a hiatus in the record. However, since detection of a hiatus is not always straightforward, we cannot fully exclude this possibility. The only alternative interpretation would be a major decrease of the reservoir effect during the deposition of the uppermost 22 cm of the sediment record. Unfortunately, we do not have robust data to support any of these possibilities so that the age-depth relation of the core remains a matter of debate. Consequently, the age of the base of the core of 17,680 ± 90 BP (Table 1) might also be questioned. Interestingly, our data largely resembles the radiocarbon dates from the previous core taken in 1996 (Valero-Garces et al., 1999) except the few hundred years lower reservoir effect of the sediment surface likely due to spatial variations within the lake basin as discussed above. However, the age model of the previous core has been re-interpreted based on three preliminary U/Th dates suggesting a much younger age of ca. 450 a BP for the base of the 1996 core (Valero-Garces et al., 2000, 2003). Accepting this age, the radiocarbon reservoir effect must have decreased in the last three centuries by about 4,000-5,000 $^{14}$C years. Due to the lack of

a critical assessment of potential bias of U/Th dating in this lake setting (Valero-Garces et al., 2000, 2003), the true age of the Peinado sediments remains unsolved. However, even if the good agreement of the radiocarbon dates in both cores does not

necessarily prove the absence of major reservoir effect changes, the postulation of a several thousand year decrease of the reservoir effect in the last few centuries should be questioned and re-investigated. Further research is needed, not only in this lake but also along the Altiplano-Puna plateau to understand how these reservoir effect changes vary across the region.

## 6 CONCLUSIONS

Radiocarbon dating of modern plants revealed large reservoir effects ranging between >12,000 and >26,000 $^{14}$C years within

the El Peinado basin. These reservoir effects result from two dominant processes: the inflow of old groundwater that likely formed during pluvial phases at the end of the last glaciation and an additional contribution of $^{14}$C-free magmatic $CO_2$. We could further prove that the reservoir effect shows distinct differences within the lake basin depending on the distance to the source of old groundwater inflow and volcanic $CO_2$. This can even influence the dating of sediment cores obtained from different locations in the lake. Through comparison of surface sediments with a previously published sediment record, we

found indications of a ca. 400-year difference in the reservoir ages between both cores. Such spatial variations in the $^{14}$C reservoir effect may also occur in other lake records in the Central Andes and elsewhere and should also be considered for the construction of age models. In this sense, corrections of $^{14}$C chronologies based on a single reservoir age for an entire lake are not reliable and would produce inaccurate chronological models, as it can vary from hundreds to thousands of years within a lake leading to temporally misleading paleoclimatic interpretations. This problem might be solved by either dating truly

terrestrial material like pollen or by applying independent dating methods like U/Th. Both, however, have also deficiencies so that constructing chronologies in environments such as that of Laguna del Peinado lake remains a major challenge. Nevertheless, the characterisation of spatial variations in reservoir effects has the potential to better assess the underlying processes influencing radiocarbon ages in a lake even if it does not fully solve the problem of reservoir effect temporal changes.

**Author contribution**

PV performed the samples processing, data analyses, and wrote the article with contributions from all co-authors. AB, FC, and RT designed the project and together with PV and CS organized fieldwork. LP helped to identify the families of the plants sampled. AB, FC and RT supervised analyses and article writing.

**Competing interests**

The authors declare that they have no conflict of interest.

## Acknowledgements

We would especially like to thank S. Vazquez and P. Martin for their invaluable help with logistics and fieldwork in this extreme region. We thank the Secretaría de Estado del Ambiente y Desarrollo Sustentable and Dirección Provincial de Biodiversidad from Gobierno de la Provincia de Catamarca (Argentina) for conceding us the working permits, and Comunidad Coya Atacameña de Antofalla for supervising fieldwork. We would also like to acknowledge Prof. Dr. Tomasz Goslar, head of the Poznan Radiocarbon Laboratory, for providing us with the necessary information on the samples treatment. We are grateful for the comments and suggestions of the two anonymous reviewers who helped us to improve this manuscript. This research was funded by the DeutscheForschungsgemeinschaft (DFG) and the Federal State of Brandenburg under the auspices of the International Research Training Group IGK2018 'SuRfAce processes, TEctonics and Georesources: The Andean foreland basin of Argentina' (STRATEGy DFG 373/34-1), and by the GFZ German Research Centre for Geosciences who provided financial support for fieldwork. This research was also carried out in the framework of the following Argentinean research projects: Agencia Nacional de Promoción Científica y Tecnológica (PICT-2019-01336), CONICET (PUE 2017-22920170100027CO and GII StRATEGy 163-A 1.1) and SECTER - Universidad Nacional de Jujuy (SeCTER-UNJu E/G011 and E/1001/-Integrar).

## Data availability

All data used in this study is available in Table 1 of the manuscript.

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
