# Peer review of "Spatial variability of the modern radiocarbon reservoir effect in the high-altitude lake Laguna del Peinado (Southern Puna Plateau, Argentina)"

_Geochronology, 2023_

## Referee Comment (RC2)

The reviewed manuscript focuses on the local variability of reservoir effects for radiocarbon dating on the Southern Puna Plateau (Argentina) comparing samples from different positions in and around Laguna del Peindado. The authors tackle reservoir effects, an important issue for the investigation of age-depth relationships for lake sediments and provide information about possible error sources.

Through the analysis of seven distinct samples, the authors demonstrate how and to which extent magmatic $CO_2$ and ancient groundwater effects the $^{14}C$ composition and deliver a detailed and well-referenced investigation of possible error sources. In addition, these results are compared to a new sediment core from the lake providing four downcore radiocarbon samples and to a core from a previous study. The overall result of the manuscript is the spatial variability of the samples around but also within the lake, opening new questions about correcting the lake's sediment chronology.

Strengths:

The aim of the manuscript is communicated clearly and the methodology of the study is well-suited to achieve it. The authors use a large number of relevant references and implement them nicely to their own results. Future studies especially at this location as well as from comparable areas will benefit from this manuscript in terms of dating. The manuscript provides all necessary information to comprehend the methods.

Weaknesses:

The overall weakness of the study is mentioned and discussed by the authors themselves: results would have benefitted from more samples, e.g. by having short cores from additional and especially more southern locations of the lake. Nevertheless, the study is well discussed and the arguments seem logic.

Contribution:

The contribution to the field seems significant, especially for studies of comparable locations. However, as the article provides no clear solution for the correction of reservoir effects for downcore samples from this lake, it only serves as a well-discussed summary and estimation about potential sources of error.

Conclusion:

Overall, this manuscript provides an interesting investigation of regional effects on radiocarbon dating and summarizes related information. It delivers a well-structured overview of previous studies from the study area and topics combined with new results. While the article has some limitations with regard to the sample size, it is a valuable addition to the literature of the studied location and to comparable areas providing insights into the interpretation of radiocarbon dates.

The manuscript follows the classical structure in an appropriate balance and is written in a clear and fluent language. I have only minor comments and questions to the manuscript:

Line 22: Please check if it would make more sense to use here the term "younger" instead of "lower".

Line 28: The introduction is very well written and the problem investigated and the aim of the study are clearly described. However, I think the manuscript might benefit from a few sentences about reservoir effects in general and/or definitions like the terms "C14-free", "C14-depleted",…. . Please consider adding some sentences.

Line 80: I am not familiar with the study area, but as it is written "currently" I asked myself if information is available about the frequency of lake level changes and/or the history of earlier connections of both lake systems. In both cases the authors should add information here.

Lines 85 – 98: The climate patterns are well described, but to follow this paragraph even better, the manuscript would benefit from an addition of the climate patterns to Fig. 1.

Page 6:

Line 139: I have three questions/comments to Table 1:

(1) I count six questions marks in the table, e.g. "Hot spring 4?". These uncertainties are not mentioned in the text or the Table caption. Question marks should be explained to avoid confusion.
(2) The first two samples result in two calibrated ages each. It should be explained why this is the case.
(3) Please explain why not all radiocarbon ages have been calibrated.

Page 9:

Line 172, 174, 180: The authors refer to Figure 4 only. Its orientation becomes clear only in comparison to Figure 1. However, I wish either an indication of e.g. "western hot spring", a north arrow or maybe a numeration of the hot springs as indicated in Fig. 1 with sample names added to Fig. 4. Otherwise, this paragraph might not be understandable without comparison to Fig. 1. Moreover, Fig. 1 should be referred in addition to Fig. 4.

Page 12:

Line 257: How do the authors proceed with the sediment core and develop the chronology? I would suggest to implement this information here or somewhere later in the manuscript.

Line 263: Are there lithological indications that would support the hypothesis of a hiatus in the sediment core?

Page 16:

Line 380-381: Please check if the published year should be changed to 2022, as indicated on the journal's homepage

---

## Author Comment (AC1)

**Response to anonymous Referee #1 comments:**

We are very grateful to Referee #1 for taking the time to carefully review our manuscript and providing positive comments that help to improve this manuscript. We provide a response to each of them below. All comments have been considered and will be included in a revised version of this manuscript.

L16-17: You obviously go on to talk about TEMPORAL variation in reservoir effects too, but I felt that this should also be mentioned right at the start here, along with your noting of "spatial variations".

Thanks for your comment, to mention temporal variations in reservoir effects we have now added "and possibly temporal" into that sentence.

L24-25: You're absolutely right about the implications of this study affecting both precision and accuracy of 14C-derived chronologies, and they're obviously interwoven, but I wonder if these should be inverted to reflect the greater importance of accuracy over precision? (I.e., accuracy is fundamental – there's no point having inaccurate chronology is there?! – and, after that, increased precision then makes the data increasingly useful, no?)

We have modified the sentence and now reads "This study has implications for accurate $^{14}$C-based chronologies and high-precision dating in paleoclimate studies in the Altiplano-Puna Plateau and similar settings".

L31: You list "endorheic basins that host numerous saline lakes, playa-lakes and salars"; is there scope for these basins to episodically dry out completely, with consequent impacts (hiatuses!) upon age modelling/palaeoenvironmental reconstruction?

Thank you for this question. Indeed, especially playa-lakes and salars can dry out episodically causing hiatuses in the sedimentary record. In order to minimize the likelihood of such hiatus, we collected the short core from Laguna del Peinado close to the deepest point of the lake at a water depth of 3.2 m. Further, we did not find any sedimentological indications in our record for a hiatus. This confirms a previous study of a sediment core that was taken even at shallower water depth (2 m) and in which no hiatus has been recognized (Valero-Garcés et al., 2000, 2003). Nevertheless, we are aware that it is difficult to detect a hiatus so that we cannot fully exclude that it has been overlooked. We address this point in the revised version of the manuscript where line 263 now reads "We do not observe lithological indications in the sediment core neither for a substantial sedimentation rate change nor for a hiatus in the record. However, since detection of a hiatus is not always straightforward, we cannot fully exclude the existence of one".

L48-49: "Sometimes, even assumptions on temporal variations of the reservoir effect are included in the construction of age-depth models"; please could you include one or two references to support this statement.

We have added references "(e.g. Grosjean et al., 2001; Moreno et al., 2007)".

L119-124: In order to interpret any radiocarbon data, it is essential to specify what chemical pre-treatment procedures have been applied. (I take on trust that this has been performed robustly, but this needs to be fully clarified, and is probably the most significant of my comments.)

Chemical pre-treatment on the samples was carried out in the Poznan Radiocarbon Laboratory where they were dated. Samples were washed with demineralised water in our lab before sending them out for dating. We have modified lines 119-121 to "Samples preparation, chemical pre-treatments, and accelerator mass spectrometry (AMS) $^{14}$C measurements were carried out in the Poznan Radiocarbon Laboratory. A full description of the procedures can be accessed at https://radiocarbon.pl/en/sample-preparation/. After mechanical removal of macroscopic contamination under binoculars, the samples underwent a sequential acid-base-acid (ABA) treatment following the protocols established for each material (UW protocol for the wood sample PEI19-P-3 and UV protocol for all other plant remains samples). Samples were first treated with 1 M HCl at 80°C for 20 min or longer if needed until gas bubbles emanations finished (UV, UW), followed by 0.1 M NaOH treatment at room temperature for fragile plant remains (UV) and 80°C for wood (UW); and then 0.25 M HCl at 80°C for 1 hr. After each treatment, samples were rinsed with deionised water (Millipore) to pH=7. The NaOH treatment step is repeated a few times until no more colouring of the solution caused by humic acids is observed. For the wood sample PEI19-P-3 (UW) an additional treatment with 5% NaClO$_2$ at room temperature was applied for 30 min."

L123-124 (and also for Table 1): Why only calibrate the post-bomb 14C measurements, but not the pre-bomb?

We do not calibrate the pre-bomb ages because the reservoir effect of these dates is not known. In order to homogenise the table, we deleted the column with calibration.

L125-127: Again, what chemical pre-treatment procedures were applied to these samples prior to d13C analysis?

We did not apply chemical pre-treatment prior to analysis in the Kiel IV Carbonate Device coupled to the IRMS. We added some information about samples and sample preparation in the methods section. Lines 125-129 now read "Additionally, $\delta^{13}C_{carb}$ was analysed in four samples from the carbonatic matrix sediments at 0-2, 24-26, 46-48, and 71-72 cm depth from the core where the plant macrofossils have been taken and in one sample from the microbial mats in the southern hot spring. Samples were freezed for 24

to 48 hours, freeze-dried for 72 hours, and ground to powder. Carbon isotopes analysis of carbonate powders ($\delta^{13}C_{carb}$) were carried out on an automated carbonate extraction device (KIEL IV) coupled to a Finnigan MAT 253 IRMS (Thermo Fisher Scientific) at the GFZ Potsdam. In brief, acid digestion of carbonates with phosphoric acid takes place in the KIEL IV to produce $CO_2$ that is ultimately analysed for $\delta^{13}C_{carb}$ in the coupled MAT 253 IRMS. Results are expressed in the conventional δ-notation in per mille (‰) relative to VPDB (Vienna Pee Dee Belemnite; Table 1). Repeated measurements of the reference material NBS 19 ensured an analytical precision better than ± 0.07 ‰ (σ)".

L129: Surely this is "precision" rather than "accuracy"?

Thanks, we have replaced "accuracy" with "precision".

L143: Your samples were collected in 2019… and so the latter age (2018-2019 cal CE) makes sense. But how do you explain the former age (1994-1996 cal CE)? A freshwater reservoir effect wouldn't ENHANCE the 14C (112.39 pMC c.f. 101.61 pMC). Precisely what was the material sampled (for both of these samples)? Is the former sample more woody material (with an associated inbuilt "storage age")? Please give more information around these samples, and suggest what has led to this.

The material collected for both samples is indeed different. The sample with an age 2018-2019 cal CE (101.61 pMC) was a Poaceae (Gramineae), possibly Festuca ortophylla (sample PEI19-P-4, Table 1, Fig. 2b). The sample with an age 1994-1996 cal CE (112.39 pMC) was a woody plant of the genus Adesmia, possibly the species horrida (sample PEI19-P-3, Table 1, Fig. 2a). Two possible causes could explain the older age (or higher pMC) of sample PEI19-P-3: 1) the plant was not alive at the time of sampling as it had no new sprouts (see Fig. 2a, the front plant was sampled); 2) the structure of this woody plant is formed over an extended time and the higher pMC than the atmosphere for 2019 results either from the integration of $^{14}C$ during the growing years or from the measurement in the sample heartwood (old wood). However, since both samples (PEI19-P-3 and PEI19-P-3) reveal modern ages and the difference between the two calibrated ages is only minor this is not relevant for the main statement that these samples are not influenced by reservoir effects. Therefore, we will not further discuss this in the revised version of the manuscript.

L166-167: I would say that this wording is misleading; Yes, terrestrial plants are "expected to provide modern radiocarbon ages without any reservoir effect involved" (generally speaking! Although there could be rare examples where the expectation may differ…) BUT aquatic plants obviously take on their carbon from the water, and so they wouldn't be "expected to" provide modern radiocarbon ages, surely? Isn't that a fundamental premise of the present paper? I just find the wording of this sentence unnecessarily misleading, taken in isolation.

We agree and have modified it to "Present-day terrestrial plants are commonly expected to provide modern radiocarbon ages, while aquatic plants potentially take up old carbon".

L168-169: This is really interesting. I am not a biologist – is the aged C being taken in by the grass from the air (localised atmospheric depletion from C release from the hydrothermal spring), or is the aged C being taken in through the roots (in the water taken up by the plant)?

Although this is difficult to ultimately prove, we assume that this plant must have absorbed aged carbon through the air by the release of $CO_{2(g)}$ from the nearby hydrothermal spring because it was not in direct contact with the hydrothermal water. Uptake of aged carbon from soil DIC could have only had a minor effect because uptake by roots commonly is 1-3% or even less of the total $CO_2$ fixed by the plant (Loczy et al., 1983; Ford et al., 2007).

L169: Clarify again that here you are referring to aquatic species(?).

We have added "aquatic" for clarification.

L182-184: Give an approximate representation of the values given for the cited study.

We have added "with differences of up to 19,000 $^{14}C$ years between different locations within individual lakes".

L191-194: "The dissolution of carbonate-rich sediments or rocks in the catchment area is usually considered a main source of 14C-dead carbon influx into a lake (Macdonald et al., 1991; Ascough et al., 2010). However, the dissolution of catchment carbonates can only be a minor source of 14C-dead carbon into Laguna del Peinado because the lithology of the basin is dominated by volcanic rocks". Does this contradict what was written earlier on ("Abundant carbonate precipitation takes place in the El Peinado basin…", L81), or do I misunderstand? (Even if the latter, perhaps clarification is still needed?)

Thank you very much for the comment that requires clarification. The main sources of water that feed Laguna del Peinado lake are the hydrothermal springs that provide the dissolved elements. As in other lakes in the Altiplano-Puna Plateau with scarce or absent carbonate outcrops in the catchment (e.g. Laguna Pastos Grandes) we interpret calcium as derived from the alteration of the volcanic bedrock by fluids at high temperatures (Muller et al., 2020). Plagioclase is dominant in the mafic rocks and ignimbrites of the El Peinado basin (Grosse et al., 2020, 2022; Kay et al., 2010). Calcium availability together with volcanic $CO_2$ supply and different processes trigger the precipitation of carbonates in these environments. In the hot springs, $CaCO_3$ precipitation is triggered by hydrothermal $CO_2$ degassing and microbially-driven elevation of local pH at crystallisation while in Laguna del Peinado, $CaCO_3$ precipitation is induced by evaporative supersaturation, $CO_2$ degassing and microbiological processes (Vignoni et al., 2022).

We have modified L81 "Carbonate precipitation takes place within both lakes and the hydrothermal springs environments as a result of hydrothermal $CO_2$ degassing, evaporation, and biological processes

with deposits comprising a wide variety of facies…". We have also added "…and extreme aridity conditions prevail" at the end of L194 and a sentence in L196 "Furthermore, calcium available for carbonate formation in this lacustrine system is interpreted to derive from the alteration of the volcanic bedrock by fluids at high temperatures as it has been observed in other systems of the Altiplano-Puna Plateau with similar characteristics (e.g. Laguna Pastos Grandes; Muller et al., 2020)".

> L213 and 216: Can you clarify what you mean by the terms "old" and "ancient" groundwater? (Is it the "100-10,000 years or longer" noted below, L219?)

This may indeed lead to confusion and we replaced "ancient" by "old" in the revised text. We have added in L214 "old waters that may be 100-10,000 yr old" and "modern water (<60 yr old)" to clarify this.

> L279: Is it possible to measure 14C on (the DIC/dissolved gasses of) the water itself? And would/could this, in combination with other isotope measures (including d13C and d3H, mentioned earlier) help to understand the "dominant process" question?

In principle, $^{14}C$ on water DIC and dissolved gasses could be measured. However, it would not help much in our case because of local dilution effects from the input of $^{14}C$-free carbon (e.g. volcanic $CO_2$) into the groundwater. Moreover, as mentioned in the previous answer, $^{3}H$ analysis would only allow us to know the proportion of modern water (<60 yr old) in the system but not the age of groundwater older than a few decades (e.g., Moran et al., preprint).

In order to attempt to resolve this issue, further multi-tracer studies would be needed to characterise the groundwater system, the processes in the recharge zones, and the volcanism of the area, which is beyond the scope of this study.

> L287: I would actually say that "corrections of 14C chronologies based on a single reservoir age for an entire lake…" would result in INACCURATE results, rather than just "large uncertainties" (which, as I noted earlier would be a bigger problem). You would only end up with "large uncertainties" if these uncertainties were ACTUALLY accounted for and, the point that I think you're making (which I totally agree with!) is that often these "large uncertainties" are NOT properly accounted for (…producing small uncertainties, but inaccurate chronologies).

Thank you very much for this comment that clarifies our statement. We have changed "large uncertainties" to "inaccurate chronological models".

> Finally, a more general question relating to your Discussion: If the C assimilated by the species in the hydrothermal pool were solely sourced from magmatic C (rather than "old groundwater"), this would yield "infinitely old" 14C ages… And so, in that scenario, even the older 14C sample would

still include some proportion of "modern" C input? (Is that reasonable to assume?) Why not perform a quick endmember "mixing model" to estimate the proportion of C (for each sample) that is from a modern (2019 CE atmospheric) source and what proportion from geologically old (14C dead) C? (N.B. this is a simple "back of an envelope" calculation, rather than requiring "proper" modelling!) I suggest that this will give a "better" impression of the differing contributions (of old vs modern C), which can be skewed by the exponential nature of the 14C decay curve, which can then carry through to all of your samples through the lake. (I.e., for each sample, what proportion of C is sourced from "modern" vs geologically "dead" sources?)

Thank you for this suggestion. We have now included a simple "mixing model" (Table 1) to assess the approximate contribution of old carbon to each sample following Pasquier-Cardin et al. (1999). We introduce this in the methods section and revised the discussion accordingly. It is true that even the sample with the oldest $^{14}$C age should include a proportion of modern carbon (~ 4% according to the model). We have now added a sentence in L233 regarding this: "Moreover, the aquatic plant with the oldest $^{14}$C age has a proportion of modern carbon (~4%; Table 1, Fig. 2d) supporting that the reservoir ages result from dilution with $^{14}$C-free volcanic $CO_2$".

(Non-comprehensive) typo/wording suggestions:

    L14: Insert comma after "This".

Thanks, we inserted the comma after "This".

    L17: Change "constrain" to "constraint".

"Constraint" has been corrected.

    L24: Here, do you mean the "centre of the lake" specifically?

We have clarified the text and specified the core location as 'in the northern part of the lake basin' and refer to Fig. 1 where the core location is shown.

    L114: Missing word: "littoral [zone]"?

We have added the missing word "zone".

    L115: Spell out "macrofossil"… Perhaps even "plant macrofossil".

We have changed "macro remain samples" to "plant macrofossil samples".

L127: "Mile" should read "mille".

Thanks, we have corrected "mile" to "mille".

L143: "cal CE" is a suffix, and so should come after the date (e.g., "1994-1996 cal CE").

Thank you for your comment. Although we no longer include calibrated dates to homogenise the data, we will consider your suggestion for future work.

L246: Even though I agree that your explanation is the overwhelmingly most likely one, is "proving" still too strong a word to use?

We agree and changed wording to "revealing".

L278: I would say that ">26,000 14C years" is more than "up to several thousand years"?!

Thanks for the comment. We have removed "up to several thousand years".

**REFERENCES**

Cartwright, I., Cendón, D., Currell, M., and Meredith, K.: A review of radioactive isotopes and other residence time tracers in understanding groundwater recharge: Possibilities, challenges, and limitations, J. Hydrol., 555, 797–811, https://doi.org/10.1016/j.jhydrol.2017.10.053, 2017.

[revised manuscript text omitted]

---

## Author Comment (AC2)

**Response to anonymous Referee #2 comments:**

We are very grateful to Referee #2 for the constructive suggestions and comments that significantly improve our manuscript. Here, we provide detailed responses to each individual comment. All comments have been considered and will be included in a revised version of this manuscript.

Line 22: Please check if it would make more sense to use here the term "younger" instead of "lower".

Thanks for your suggestion. We have changed "lower" to "younger".

Line 28: The introduction is very well written and the problem investigated and the aim of the study are clearly described. However, I think the manuscript might benefit from a few sentences about reservoir effects in general and/or definitions like the terms "C14-free", "C14-depleted",…. . Please consider adding some sentences.

Thanks for your suggestion. We have now included a few sentences in the introduction.

Line 36 now reads "Our understanding of the regional and temporal hydroclimatic dynamics in the Altiplano-Puna Plateau is hampered by the difficulty in obtaining accurate chronologies from lacustrine sediments due to the scarcity of terrestrial organic matter and the anomalously old apparent $^{14}$C age of waters and hence aquatic samples, known as "reservoir effect" (Grosjean et al., 1995, 1997, 2001; Geyh et al., 1998; Valero-Garcés et al., 2000; Yu et al., 2007)".

We have also modified line 42 and now reads "Therefore, obtaining reliable chronological models using lake sediments from this region is critical and requires an understanding of the $^{14}$C reservoir effect variability in each particular lake system as it depends on the $CO_2$ exchange rate between the water and the atmosphere, the internal system mixing dynamics, and the input of $^{14}$C-dead (i.e. derived from carbonates), $^{14}$C-depleted (i.e. dilution of the initial $^{14}$C content), or $^{14}$C-free carbon (e.g. volcanic $CO_2$; Macdonald et al., 1991; Ascough et al., 2010; Keaveney and Reimer, 2012; Jull et al., 2013; Lockot et al., 2015)".

Line 80: I am not familiar with the study area, but as it is written "currently" I asked myself if information is available about the frequency of lake level changes and/or the history of earlier connections of both lake systems. In both cases the authors should add information here.

Unfortunately, there is no information on the frequency of lake level changes in these lakes. The only information on the history of past connections between both lakes comes from recent satellite images showing a connection until ca. 2005 (Villafañe et al., 2021).

We have added a comment on this in line 80: "Both lakes were connected until ca. 2005 according to satellite images (Villafañe et al., 2021)", and line 250: "…probably related to a lake level lowering of at least 0.6 m with the consequent disconnection between Laguna del Peinado and Laguna Turquesa (Villafañe et al., 2021)".

Lines 85 – 98: The climate patterns are well described, but to follow this paragraph even better, the manuscript would benefit from an addition of the climate patterns to Fig. 1.

Thanks for your suggestion. We have modified Fig. 1 to include the climate patterns.

[Figure]

Figure 1: Location and and type of samples collected in the El Peinado basin during 2019 (© Google Earth 2020, Maxar Technologies, CNES/Airbus). Sediment core samples are indicated in italics. Left top corner: map of South America with the Altiplano-Puna Plateau highlighted in brown and the climatic moisture sources (SAMS-South American Monsoon System and

SHPW-Southern Hemisphere Pacific Westerlies). The red square marks the approximate location of the El Peinado basin in the Puna Plateau of NW Argentina.

Page 6:

Line 139: I have three questions/comments to Table 1:

- I count six questions marks in the table, e.g. "Hot spring 4?". These uncertainties are not mentioned in the text or the Table caption. Question marks should be explained to avoid confusion.

Thanks for your comment. We have now added an explanation of the question marks in the Table caption "Question marks (?) denote samples where water influence, water mixing, and plants genus and/or species could not be determined with certainty". The question mark in "Hot spring 4 (shallow, mix with lake water?)" is indeed discussed in the text in line 180-181: "The difference of ca. 8,000 $^{14}$C years between both hot springs (Table 1, Fig. 1 and 4) could result either from some influence (i.e. mixing) of lake water with higher $^{14}$C concentrations in the western shore hot spring (Fig. 1), or from the existence of separate hydrothermal systems bounded by the Peinado lineament with distinct $^{14}$C content in the DIC". For the question mark in "Hot spring 4?" we have added a brief discussion in line 239: "This might explain the old age of the terrestrial plant sample since it grew at a distance of only ~15 cm from the local hot spring. Potential uptake of soil DIC through the roots might additionally contribute but only to a very minor degree since it usually represents less than 1% of the total $CO_2$ fixed by the plant and may be a more relevant source of carbon for the underground tissues (Loczy et al., 1983; Brix, 1990; Enoch and Olesen, 1993; Ford et al., 2007)".

- The first two samples result in two calibrated ages each. It should be explained why this is the case.

In order to report ages in a consistent way, we resign from calibrating the modern ages. A calibration of only these two ages further distracts from the main focus of the study (see also comment to reviewer 1)

- Please explain why not all radiocarbon ages have been calibrated.

Thank you for pointing this out. It does not make sense to calibrate only two ages since calibrating the other data is not useful without known reservoir ages. Therefore, we resign from calibrating only the two modern samples (see comment above and to reviewer 1).

Page 9:

Line 172, 174, 180: The authors refer to Figure 4 only. Its orientation becomes clear only in comparison to Figure 1. However, I wish either an indication of e.g. "western hot spring", a north arrow or maybe a numeration of the hot springs as indicated in Fig. 1 with sample names added to

Fig. 4. Otherwise, this paragraph might not be understandable without comparison to Fig. 1. Moreover, Fig. 1 should be referred in addition to Fig. 4.

Thank you for your suggestion. We have modified Fig. 4 and included the names of the samples as well as an arrow indicating north. We now also refer to Fig. 1.

[Figure]

Figure 4: Aerial view of Laguna del Peinado from the northeast and all radiocarbon dates obtained from modern surface samples. For a top view, please refer to Figure 1.

Page 12:

Line 257: How do the authors proceed with the sediment core and develop the chronology? I would suggest to implement this information here or somewhere later in the manuscript.

We have now included in the conclusions of the manuscript (line 289) information on how we will proceed with the chronology of the sediment cores: "In contrast to proving spatial variability of reservoir ages, it remains challenging to determine temporal changes in reservoir effects in absence of robust independent dating methods. One potential option would be radiocarbon dating on pollen which, however, failed due to the scarcity of vegetation in the region. Another option is U/Th dating which is currently evaluated. However, the sedimentary environment of the Peinado lake is challenging for this method as well. Therefore, the potential of radiocarbon dating of lake sediment cores from the Central Andes is limited and remains a major challenge."

Line 263: Are there lithological indications that would support the hypothesis of a hiatus in the sediment core?

We have added the following information in line 263: "We do not observe lithological indications in the sediment core neither for a substantial sedimentation rate change nor for a hiatus in the record. However, since detection of a hiatus is not always straightforward, we cannot fully exclude the existence of one". Please see also the comment to reviewer 1 regarding line 31.

Page 16:

Line 380-381: Please check if the published year should be changed to 2022, as indicated on the journal's homepage

Correct, we have changed the publication year to 2022.

**REFERENCES**

Ascough, P. L., Cook, G. T., Church, M. J., Dunbar, E., Einarsson, Á., McGovern, T. H., Dugmore, A. J., Perdikaris, S., Hastie, H., Friðriksson, A., and Gestsdóttir, H.: Temporal and Spatial Variations in Freshwater 14 C Reservoir Effects: Lake Mývatn, Northern Iceland, Radiocarbon, 52, 1098–1112, https://doi.org/10.1017/S003382220004618X, 2010.

Brix, H.: Uptake and photosynthetic utilization of sediment-derived carbon by Phragmites australis (Cav.) Trin. ex Steudel, Aquat. Bot., 38, 377–389, https://doi.org/10.1016/0304-3770(90)90032-G, 1990.

Enoch, H. Z. and Olesen, J. M.: Plant response to irrigation with water enriched with carbon dioxide, New Phytol., 125, 249–258, https://doi.org/10.1111/j.1469-8137.1993.tb03880.x, 1993.

Ford, C. R., Wurzburger, N., Hendrick, R. L., and Teskey, R. O.: Soil DIC uptake and fixation in Pinus taeda seedlings and its C contribution to plant tissues and ectomycorrhizal fungi, Tree Physiol., 27, 375–383, https://doi.org/10.1093/treephys/27.3.375, 2007.

Geyh, M. A., Schotterer, U., and Grosjean, M.: Temporal Changes of the 14 C Reservoir Effect in Lakes, Radiocarbon, 40, 921–931, https://doi.org/10.1017/S0033822200018890, 1998.

Grosjean, M., Geyh, M., Messerli, B., and Schotterer, U.: Late-glacial and early Holocene lake sediments, groundwater formation and climate in the Atacama Altiplano 22-24 S, J. Paleolimnol., 14, 241–252, 1995.

Grosjean, M., Valero-Garcés, B. L., Geyh, M. A., Messerli, B., Schotterer, U., Schreier, H., and Kelts, K.: Mid- and late-Holocene limnogeology of Laguna del Negro Francisco, northern Chile, and its palaeoclimatic implications, 7, 151–159, https://doi.org/10.1177/095968369700700203, 1997.

Grosjean, M., van Leeuwen, J. F. N., van Der Knaap, W. O., Geyh, M. A., Ammann, B., Tanner, W., Messerli, B., Núñez, L. A., Valero-Garcés, B. L., and Veit, H.: A 22,000 14C year BP sediment and pollen record of climate change from Laguna Miscanti (23°S), northern Chile, Glob. Planet. Change, 28, 35–51, https://doi.org/10.1016/S0921-8181(00)00063-1, 2001.

Jull, A. J. T., Burr, G. S., and Hodgins, G. W. L.: Radiocarbon dating, reservoir effects, and calibration, Quat. Int., 299, 64–71, https://doi.org/10.1016/j.quaint.2012.10.028, 2013.

Keaveney, E. M. and Reimer, P. J.: Understanding the variability in freshwater radiocarbon reservoir offsets: A cautionary tale, J. Archaeol. Sci., 39, 1306–1316, https://doi.org/10.1016/j.jas.2011.12.025, 2012.

Lockot, G., Ramisch, A., Wünnemann, B., Hartmann, K., Haberzettl, T., Chen, H., and Diekmann, B.: A Process- and Provenance-Based Attempt to Unravel Inconsistent Radiocarbon Chronologies in Lake Sediments: An Example from Lake Heihai, North Tibetan Plateau (China), Radiocarbon, 57, 1003–1019, https://doi.org/10.2458/azu_rc.57.18221, 2015.

Loczy, S., Carignan, R., and Planas, D.: The role of roots in carbon uptake by the submersed macrophytes Myriophyllum spicatum, Vallisneria americana, and Heteranthera dubia, Hydrobiologia, 98, 3–7, https://doi.org/10.1007/BF00019244, 1983.

Macdonald, A. G. M., Beukens, R. P., and Kieser, W. E.: Radiocarbon Dating of Limnic Sediments: A Comparative Analysis and Discussion, Ecology, 72, 1150–1155, 1991.

Valero-Garcés, B. L., Delgado-Huertas, A., Ratto, N., Navas, A., and Edwards, L.: Paleohydrology of Andean saline lakes from sedimentological and isotopic records, Northwestern Argentina, J. Paleolimnol., 24, 343–359, https://doi.org/10.1023/A:1008146122074, 2000.

Villafañe, P. G., Cónsole-Gonella, C., Cury, L. F., and Farías, M. E.: Short-term microbialite resurgence as indicator of ecological resilience against crises (Catamarca, Argentine Puna), Environ. Microbiol. Rep., 13, 659–667, https://doi.org/10.1111/1758-2229.12977, 2021.

Yu, S.-Y., Shen, J., and Colman, S. M.: Modeling the Radiocarbon Reservoir Effect in Lacustrine Systems, Radiocarbon, 49, 1241–1254, https://doi.org/10.1017/S0033822200043150, 2007.

---

## Author Response (AR1)

**Author's response**

P. A. Vignoni, F. E. Córdoba, R. Tjallingii, C. Santamans, L. C. Lupo, A. Brauer

We are very grateful to Referee #1, Referee #2, and the Associate Editor Irka Hajdas for carefully reviewing our manuscript and providing constructive suggestions and comments that significantly improved it. Here, we include the reviews received for this manuscript and provide detailed responses indicating the changes made to the text. All comments have been considered in the revised version of this manuscript.

**Referee #1:**

**L16-17: You obviously go on to talk about TEMPORAL variation in reservoir effects too, but I felt that this should also be mentioned right at the start here, along with your noting of "spatial variations".**

We have added "and possibly temporal variations" into that sentence. We have also added a new sentence on line 25 of the revised text to stress that temporal changes are probable: "Temporal changes of reservoir effects in sediment records are more difficult to quantify but $^{14}$C ages from a short core from Laguna del Peinado may suggest temporal reservoir age variations of a few thousand years".

**L24-25: You're absolutely right about the implications of this study affecting both precision and accuracy of 14C-derived chronologies, and they're obviously interwoven, but I wonder if these should be inverted to reflect the greater importance of accuracy over precision? (I.e., accuracy is fundamental – there's no point having inaccurate chronology is there?! – and, after that, increased precision then makes the data increasingly useful, no?)**

We have modified the sentence and line 26-27 of the revised manuscript now reads "This study has implications for accurate $^{14}$C-based chronologies for paleoclimate studies in the Altiplano-Puna Plateau and similar settings".

**L31: You list "endorheic basins that host numerous saline lakes, playa-lakes and salars"; is there scope for these basins to episodically dry out completely, with consequent impacts (hiatuses!) upon age modelling/palaeoenvironmental reconstruction?**

We address this point in line 291 of the revised version of the manuscript. We have added: "We do not observe lithological indications in the sediment core neither for a substantial sedimentation rate change nor for a hiatus in the record. However, since detection of a hiatus is not always straightforward, we cannot fully exclude this possibility".

**L48-49: "Sometimes, even assumptions on temporal variations of the reservoir effect are included in the construction of age-depth models"; please could you include one or two references to support this statement.**

We have added the references in line 54 of the revised manuscript: "(e.g. Grosjean et al., 2001; Moreno et al., 2007)".

**L119-124: In order to interpret any radiocarbon data, it is essential to specify what chemical pre-treatment procedures have been applied. (I take on trust that this has been performed robustly, but this needs to be fully clarified, and is probably the most significant of my comments.)**

We have now specified the chemical pre-treatment procedures that were applied. The new paragraph on radiocarbon analysis (lines 122-136) in the Materials and Methods section reads as follows:

"Samples preparation, chemical pre-treatments, and accelerator mass spectrometry (AMS) $^{14}$C measurements were carried out in the Poznan Radiocarbon Laboratory (Poland). A full description of the procedures can be accessed at: https://radiocarbon.pl/en/sample-preparation/. After mechanical removal of macroscopic contamination under binoculars, the samples underwent a sequential acid-base-acid (ABA) treatment following the protocols established for each material (UW protocol for the wood sample PEI19-P-3 and UV protocol for all other plant samples). Samples were first treated with 1 M HCl at 80°C for 20 min or longer if needed until gas bubbles emanations finished (UV, UW), followed by 0.1M NaOH treatment at room temperature for fragile plant remains (UV) and 80°C for wood (UW); and then 0.25M HCl at 80°C for 1 hr. After each treatment, samples were rinsed with deionised water (Millipore) to pH=7. The NaOH treatment step is repeated a few times until no more colouring of the solution caused by humic acids is observed. For the wood sample PEI19-P-3 (UW), an additional treatment with 5% NaClO2 at room temperature was applied for 30 min. The resulting $^{14}$C ages are listed in Table 1 with one standard deviation (σ). Samples with percentage of modern carbon (pMC) and radiocarbon ages were converted to fraction modern $^{14}$C ($F^{14}$C) values (Stuiver and Polach, 1977; Reimer et al., 2004; Stenström et al., 2011) using the R package 'rintcal' (Blaauw, 2003). Two post-bomb dates from terrestrial plant samples collected at the shore without contact to lake water were calibrated with CALIBomb (Reimer et al., 2004; Reimer and Reimer, 2023) using the Southern Hemisphere Zone 1-2 calibration data set (Hua et al., 2022)."

**L123-124 (and also for Table 1): Why only calibrate the post-bomb 14C measurements, but not the pre-bomb?**

We did not calibrate the pre-bomb ages because the reservoir effect of these dates is not known. We now clarify on line 134 that we only calibrate the two post-bomb ages (see answer above). We consider the calibration as not relevant for our main argumentation and, therefore, deleted the column with calibrated ages from Table 1 but kept it in the text as an additional information (lines 154-155, 188-191 of the revised text).

**L125-127: Again, what chemical pre-treatment procedures were applied to these samples prior to d13C analysis?**

We did not apply chemical pre-treatment prior to analysis. However, we have added further information on the samples and their processing. The new paragraph on carbon isotopes analysis (lines 143-150) in the Materials and Methods section reads as follows:

"Additionally, $\delta^{13}C_{carb}$ was analysed in four samples from the carbonatic matrix sediments at 0-2, 24-26, 46-48, and 71-72 cm depth from the core where the plant macrofossils have been taken, and in one sample from the microbial mats in the southern hot spring. Samples were frozen for 24 to 48 hours, freeze-dried for 72 hours, and ground to powder. Carbon isotopes analysis of carbonate powders ($\delta^{13}C_{carb}$) were carried out on an automated carbonate extraction device (KIEL IV) coupled to a Finnigan MAT 253 IRMS (Thermo Fisher Scientific) at the GFZ Potsdam. In brief, acid digestion of carbonates with phosphoric acid takes place in the KIEL IV to produce $CO_2$ that is ultimately analysed for $\delta^{13}C_{carb}$ in the coupled MAT 253 IRMS. Results are expressed in the conventional $\delta$-notation in per mille (‰) relative to VPDB (Vienna Pee Dee Belemnite; Table 1). Repeated measurements of the reference material NBS 19 ensured an analytical precision better than $\pm 0.07$ ‰ ($\sigma$)."

**L129: Surely this is "precision" rather than "accuracy"?**

We have replaced "accuracy" with "precision" (now line 150).

**L143: Your samples were collected in 2019… and so the latter age (2018-2019 cal CE) makes sense. But how do you explain the former age (1994-1996 cal CE)? A freshwater reservoir effect wouldn't ENHANCE the 14C (112.39 pMC c.f. 101.61 pMC). Precisely what was the material sampled (for both of these samples)? Is the former sample more woody material (with an associated inbuilt "storage age")? Please give more information around these samples, and suggest what has led to this.**

We believe the most likely explanation for the 1994-1996 cal CE age obtained for sample PEI19-P-3 is that the plant was not alive at the time of sampling as it had no new sprouts. We now explain it in the text in lines 187-191:

 "The only two terrestrial plant samples in our study that yielded recent ages and were free of any reservoir effects were found 15 and 5 m away from the lake shore line (Fig. 1 and 4). PEI19-P-3, a woody plant of the genus *Adesmia,* was most likely dead at the time of sampling as it had no sprouts explaining the 1994-1996 cal CE age (Table 1, Fig. 2a, the front plant was sampled) while PEI19-P-4, a Poaceae possibly *Festuca ortophylla*, dated to 2018-2019 cal CE in agreement with the sampling year (Table 1, Fig. 2b)."

However, as noted in our previous response, the relevant point for this study is that both samples show modern ages and neither is affected by reservoir effects.

**L166-167: I would say that this wording is misleading; Yes, terrestrial plants are "expected to provide modern radiocarbon ages without any reservoir effect involved" (generally speaking! Although there could be rare examples where the expectation may differ…) BUT aquatic plants obviously take on their carbon from the water, and so they wouldn't be "expected to" provide modern radiocarbon ages, surely? Isn't that a fundamental premise of the present paper? I just find the wording of this sentence unnecessarily misleading, taken in isolation.**

We agree and have modified it to "Present-day terrestrial plants are commonly expected to provide modern radiocarbon ages, while aquatic plants potentially take up old carbon" (now line 186).

**L168-169: This is really interesting. I am not a biologist – is the aged C being taken in by the grass from the air (localised atmospheric depletion from C release from the hydrothermal spring), or is the aged C being taken in through the roots (in the water taken up by the plant)?**

In order to clarify we have added a sentence and lines 263-268 of the revised manuscript read as follows:

"It has been reported that diffuse emanations of magmatic $CO_2$ through soils lead to a substantial $^{14}C$ depletion in terrestrial plants when $^{14}C$-free $CO_2$ is assimilated during photosynthesis (Pasquier-Cardin et al., 1999). This might explain the old age of the terrestrial plant sample since it grew at a distance of only ~15 cm from the local hot spring and was not in direct contact with water. Potential uptake of soil DIC through the roots might additionally contribute but only to a very minor degree since it usually represents less than 1% of the total $CO_2$ fixed by plants (Loczy et al., 1983; Brix, 1990; Enoch and Olesen, 1993; Ford et al., 2007)."

**L169: Clarify again that here you are referring to aquatic species(?).**

We have added "aquatic" for clarification (now line 193).

**L182-184: Give an approximate representation of the values given for the cited study.**

We now give the values for the cited study. Lines 206-209 of the revised text read as follows:

"A similar pattern of spatial variability has been observed in lacustrine systems in the Tibetan Plateau, with high reservoir effect in tributaries and spring waters and lower reservoir effect in the central regions of lakes with differences of up to 19,000 $^{14}C$ years between different locations within individual systems (Mischke et al., 2013)."

**L191-194: "The dissolution of carbonate-rich sediments or rocks in the catchment area is usually considered a main source of 14C-dead carbon influx into a lake (Macdonald et al., 1991; Ascough et al., 2010). However, the dissolution of catchment carbonates can only be a minor source of 14C-dead carbon into Laguna del Peinado because the lithology of the basin is dominated by volcanic rocks". Does this contradict what was written earlier on ("Abundant carbonate precipitation takes place in the El Peinado basin…", L81), or do I misunderstand? (Even if the latter, perhaps clarification is still needed?)**

We agree with the reviewer that clarification is needed and therefore we have made some changes to the text. We have modified line 85 (former line 81) and now reads "Carbonate precipitation takes place within both lakes and the hydrothermal springs environments as a result of $CO_2$ degassing, evaporation, and biological processes…". At the end of line 220 we have also added "…and extreme arid conditions prevail". Finally, we have included a statement in line 237 that reads as follow: "Furthermore, calcium available for carbonate formation in this lacustrine system is interpreted to derive from the alteration of the volcanic bedrock by fluids at high temperatures as described in other lake systems in the Altiplano-Puna Plateau (e.g. Laguna Pastos Grandes; Muller et al., 2020)".

**L213 and 216: Can you clarify what you mean by the terms "old" and "ancient" groundwater? (Is it the "100-10,000 years or longer" noted below, L219?)**

To avoid confusion, in the revised manuscript we have changed the word "ancient" to "old" in lines 224 and 242. We have also added a statement to clarify and lines 238-240 now read:

"The influence of old groundwater in the lake is consistent with $^3H$ analysis in wetland systems in the Southern Puna Plateau proving that these environments are mainly sustained by old waters that can be centuries to several millenia old, with only minor contribution of modern water (<60 years old) not exceeding 10% (Moran et al., 2019, 2021; Frau et al., 2021)".

**L279: Is it possible to measure 14C on (the DIC/dissolved gasses of) the water itself? And would/could this, in combination with other isotope measures (including d13C and d3H, mentioned earlier) help to understand the "dominant process" question?**

This is a very interesting question, however, this kind of measurements would not help much in our study area because of the dilution effect from the input of $^{14}C$-free volcanic $CO_2$. We mention an example of $^{14}C$ measurements on the DIC in both the original and revised text (lines 231 and 258, respectively). We have added in line 244 the following: "Although $^3H$ data of waters is lacking for the El Peinado basin, this system most likely is almost entirely supported by old groundwater as reported also from other sites in the region (e.g. Frau et al., 2021; Moran et al., 2019, 2021; Godfrey et al., 2021)."

**L287: I would actually say that "corrections of 14C chronologies based on a single reservoir age for an entire lake…" would result in INACCURATE results, rather than just "large uncertainties" (which, as I noted earlier would be a bigger problem). You would only end up with "large uncertainties" if these uncertainties were ACTUALLY accounted for and, the point that I think you're making (which I totally agree with!) is that often these "large uncertainties" are NOT properly accounted for (…producing small uncertainties, but inaccurate chronologies).**

We have changed "large uncertainties" to "inaccurate chronological models" in line 317 of the revised manuscript.

**Finally, a more general question relating to your Discussion: If the C assimilated by the species in the hydrothermal pool were solely sourced from magmatic C (rather than "old groundwater"), this would yield "infinitely old" 14C ages… And so, in that scenario, even the older 14C sample would still include some proportion of "modern" C input? (Is that reasonable to assume?) Why not perform a quick endmember "mixing model" to estimate the proportion of C (for each sample) that is from a modern (2019 CE atmospheric) source and what proportion from geologically old (14C dead) C? (N.B. this is a simple "back of an envelope" calculation, rather than requiring "proper" modelling!) I suggest that this will give a "better" impression of the differing contributions (of old vs modern C), which can be skewed by the exponential nature of the 14C decay curve, which can then carry through to all of your samples through the lake. (I.e., for each sample, what proportion of C is sourced from "modern" vs geologically "dead" sources?)**

As suggested, we have included the simple "mixing model" in the revised manuscript. This is now included in the methods section in lines 137-142:

"We conducted a simple end-member mixing model to calculate the approximate proportion of dead ($^{14}$C-free) versus modern (i.e. atmospheric) carbon in each sample following Pasquier-Cardin et al. (1999) as:

$$Dead\ carbon\ (\%) = [1 - (F^{14}C\ in\ sample/F^{14}C\ in\ reference\ plant)] \times 100 \quad (1)$$

We considered sample PEI19-P-4 as the reference plant that best represents local atmospheric $F^{14}$C (Table 1) at the time of sampling (2019) compared to the average value for Southern Hemisphere Zone 1-2 (1.019; Hua et al., 2022). We assumed that the $^{14}$C content in this sample was in equilibrium with the local atmospheric carbon."

The conversion of radiocarbon ages to $F^{14}$C is detailed in line 132 of the modified paragraph about radiocarbon analysis described in a previous answer.

We have modified Table 1 and included two columns with the "$F^{14}$C" values and the "Proportion of dead carbon". To show these results more graphically, we have also added pie charts in Figure 2 showing the estimated proportion of modern and dead carbon for each sample. For Figure 2 and Table 1 please refer to the revised version of the manuscript.

The results and discussion sections have been also revised accordingly. We have added the following:

Lines 163-165: "A simple mixing model revealed highest proportion of dead carbon in microbial mats from the southern and western hot springs (96.4% and 90.2%, respectively), while values for the lake modern aquatic macrophytes ranged between ~78 and 82% (Table 1, Fig. 2)."

Lines 191-193: "Another Poaceae sample (PEI19-HTS4-T-1) growing in the vicinity of a hydrothermal spring (~15 cm) revealed an age of $1,580 \pm 30$ BP indicating incorporation of $^{14}$C-depleted carbon (~20%; Table 1, Fig. 2d)."

Lines 260-261: "Moreover, the aquatic plant with the oldest $^{14}$C age has a proportion of modern carbon (~ 4%; Table 1, Fig. 2d) supporting that the reservoir ages result from dilution with $^{14}$C-free volcanic $CO_2$."

**(Non-comprehensive) typo/wording suggestions:**

**L14: Insert comma after "This".**

We have inserted the comma after "This" (line 14).

**L17: Change "constrain" to "constraint".**

"Constraint" has been corrected (line 17).

**L24: Here, do you mean the "centre of the lake" specifically?**

We have clarified in the revised text. Lines 23-24 now read: "Altogether, our findings reveal a spatial variability of up to 14,000 $^{14}$C years of the modern reservoir effect between the hot springs and the northern part of the Peinado lake basin".

**L114: Missing word: "littoral [zone]"?**

We have added the missing word "zone" (line 117 of the revised text).

**L115: Spell out "macrofossil"… Perhaps even "plant macrofossil".**

We now refer to "plant macrofossil samples" (line 118).

**L127: "Mile" should read "mille".**

This has been changed (line 149).

**L143: "cal CE" is a suffix, and so should come after the date (e.g., "1994-1996 cal CE").**

We have modified the text accordingly and in the revised version "cal CE" is placed after the date (lines 154, 155, 189, and 190).

**L246: Even though I agree that your explanation is the overwhelmingly most likely one, is "proving" still too strong a word to use?**

We agree with the reviewer and changed wording to "revealing" (line 275).

**L278: I would say that ">26,000 14C years" is more than "up to several thousand years"?!**

It is indeed. We have modified line 308 (former line 278) and now reads as follows: "Radiocarbon dating of modern plants revealed large reservoir effects ranging between >12,000 and >26,000 $^{14}$C years within the El Peinado basin."

**Referee #2**

**Page 1**

**Line 22: Please check if it would make more sense to use here the term "younger" instead of "lower".**

We have changed "lower" to "younger" (line 22).

**Line 28: The introduction is very well written and the problem investigated and the aim of the study are clearly described. However, I think the manuscript might benefit from a few sentences about reservoir effects in general and/or definitions like the terms "C14-free", "C14-depleted",…. . Please consider adding some sentences.**

As suggested, we have revised the Introduction. Line 38 of the revised text now reads:

"Our understanding of the regional and temporal hydroclimatic dynamics in the Altiplano-Puna Plateau is hampered by the difficulty in obtaining accurate chronologies from lacustrine sediments due to the scarcity of terrestrial organic matter and the anomalously old apparent $^{14}$C age of waters and hence aquatic samples, known as "reservoir effect" (Grosjean et al., 1995, 1997, 2001; Geyh et al., 1998; Valero-Garcés et al., 2000; Yu et al., 2007)."

We have also added a sentence in line 46:

"Reservoir effects depend on different causes including $CO_2$ exchange rates between the water and the atmosphere, the internal system mixing dynamics, and the input of $^{14}$C-free ('dead') or $^{14}$C-depleted carbon

either derived from dissolved carbonates, volcanic $CO_2$ or the inflow of old groundwater (Macdonald et al., 1991; Ascough et al., 2010; Keaveney and Reimer, 2012; Jull et al., 2013; Lockot et al., 2015)."

**Page 3**

**Line 80: I am not familiar with the study area, but as it is written "currently" I asked myself if information is available about the frequency of lake level changes and/or the history of earlier connections of both lake systems. In both cases the authors should add information here.**

We have added a sentence in line 84 (former line 80): "Both lakes were connected until ca. 2005 according to satellite images (Villafañe et al., 2021)". We have also included a comment on this in line 278 (former line 250): "…probably related to a lake level lowering of at least 0.6 m and the associated disconnection between Laguna del Peinado and Laguna Turquesa (Villafañe et al., 2021)".

**Lines 85 – 98: The climate patterns are well described, but to follow this paragraph even better, the manuscript would benefit from an addition of the climate patterns to Fig. 1.**

We have revised Figure 1 and it now includes a map with the climatic moisture sources. Please refer to Figure 1 of the revised manuscript.

**Page 6:**

**Line 139: I have three questions/comments to Table 1:**

- **I count six questions marks in the table, e.g. "Hot spring 4?". These uncertainties are not mentioned in the text or the Table caption. Question marks should be explained to avoid confusion.**

We have modified Table 1 caption. In the revised manuscript (line 170) it reads as follows:

"Table 1: $^{14}$C ages from El Peinado basin. $F^{14}$C values were calculated with the package 'rintcal' (Blaauw, 2003). The proportion of dead ($^{14}$C-free) carbon was calculated with reference to sample PEI19-P-4, considered representative the local atmospheric $F^{14}$C. As a reference, for the year 2019 when these samples were collected, the mean value of atmospheric $F^{14}$C for the Southern Hemisphere Zone 1-2 is 1.019 (January to May; Hua et al., 2022). The $\delta^{13}$C values in italic correspond to samples at 24 to 26 cm and 46 to 48 cm, and differ at the sampling depths for $^{14}$C. Question marks (?) denote samples where water influence, water mixing, and plants genus and/or species could not be determined with certainty."

As clarified in the "Reply to Referee #2" file, some of the question marks are indeed discussed in both the original and revised manuscript (lines 204-206 and 264-268 of the revised text).

- **The first two samples result in two calibrated ages each. It should be explained why this is the case.**
- **Please explain why not all radiocarbon ages have been calibrated.**

In the revised manuscript we have deleted calibration from Table 1 because it is not essential for the discussion and distracts from the main focus of the study (see also comment to Referee #1).

**Page 9:**

**Line 172, 174, 180: The authors refer to Figure 4 only. Its orientation becomes clear only in comparison to Figure 1. However, I wish either an indication of e.g. "western hot spring", a north arrow or maybe a numeration of the hot springs as indicated in Fig. 1 with sample names added to Fig. 4. Otherwise, this paragraph might not be understandable without comparison to Fig. 1. Moreover, Fig. 1 should be referred in addition to Fig. 4.**

We have modified Fig. 4 and included the names of the samples as well as an arrow indicating north. We now also refer to Fig. 1 (lines 188, 196, 198, 202, and 206).

**Page 12:**

**Line 257: How do the authors proceed with the sediment core and develop the chronology? I would suggest to implement this information here or somewhere later in the manuscript.**

We have included this information in the Conclusions of the revised manuscript in line 318:

"This problem might be solved by either dating truly terrestrial material like pollen or by applying independent dating methods like U/Th. Both, however, have also deficiencies so that constructing chronologies in environments such as that of Laguna del Peinado lake remains a major challenge. Nevertheless, the characterisation of spatial variations in reservoir effects has the potential to better assess the underlying processes influencing radiocarbon ages in a lake even if it does not fully solve the problem of reservoir effect temporal changes."

**Line 263: Are there lithological indications that would support the hypothesis of a hiatus in the sediment core?**

We have added a comment in line 291 (former line 263) of the revised manuscript: "We do not observe lithological indications in the sediment core neither for a substantial sedimentation rate change nor for a hiatus in the record. However, since detection of a hiatus is not always straightforward, we cannot fully exclude this possibility."

**Page 16:**

**Line 380-381: Please check if the published year should be changed to 2022, as indicated on the journal's homepage**

It is now corrected to 2022.

**Other changes made by the authors**

**Line 13:** We have replaced "often" with "commonly".

**Line 41:** We have added the reference "Yu et al., 2007".

**Line 91 and 97:** We have added "; Fig. 1".

**Line 104:** We have modified the title of Figure 1 based on the suggested changes. It now reads: "Figure 1: Location and type of samples collected in the El Peinado basin during 2019 (© Google Earth 2020, Maxar Technologies, CNES/Airbus). Sediment core samples are indicated in italics. Left top corner: map of South America with the Altiplano-Puna Plateau highlighted in brown and the climatic moisture sources (SAMS: South American Monsoon System, SHPW: Southern Hemisphere Pacific Westerlies). The red square marks the approximate location of the El Peinado basin in the Puna Plateau of NW Argentina."

**Line 111:** We added "/modern".

**Line 153:** We added "away".

**Line 176-Figure 2:** We have modified the lettering (or numbering) of the images within the figure to separate two samples that were previously indicated together (now 'c' and 'd'). We have also added in the bottom corners of each image a pie chart in reference to the result of the "mixing model" suggested by Referee #1. Therefore, we have made the necessary changes in the text with reference to the new order of the images (lines 156, 158, 160, and 193). The modified figure caption now reads as follows: "Figure 2: Modern samples: (a) and (b) terrestrial, (c) aquatic and (d) terrestrial by the western shore hydrothermal spring, (e) aquatic from the southern shore hydrothermal spring, (f) lake littoral, (g) aquatic from the top of the lake short core. The pie charts in the bottom corners show the estimated proportion of modern and dead carbon for each sample (Table 1)."

**Line 179-Figure 3:** We no longer include the sedimentation rates.

**Line 183:** We modified the first sentence and moved it to the Introduction to include there information on the reservoir effect in general as suggested by Referee #2 (see comment above). We also replaced "that lead to the" by "$^{14}$C-free or $^{14}$C-depleted carbon causing", and added an "s" at the end of reservoir effect.

**Line 210-Figure 4:** We have modified the figure as suggested. The figure caption now reads: "Figure 4: Aerial view of Laguna del Peinado from the northeast and all radiocarbon dates obtained from modern surface samples. For a top view, please refer to Figure 1."

**Line 233:** We replaced "dead" by "free".

**Line 234:** We added "Fig. 1".

**Line 270:** We have added "Fig. 2a, 2b, and 4".

**Line 301:** We have replaced "age" by "effect".

**Lines 333-335:** We have added in the Acknowledgements section the following: "We would also like to acknowledge Prof. Dr. Tomasz Goslar, head of the Poznan Radiocarbon Laboratory, for providing us with the necessary information on the samples treatment. We are grateful for the comments and suggestions of the two anonymous reviewers who helped us to improve this manuscript".

**Line 342:** We have added "and E/1001/-Integrar".

**REFERENCES**

[revised manuscript text omitted]

---

## Author Response (AR2)

**Author's response**

P. A. Vignoni, F. E. Córdoba, R. Tjallingii, C. Santamans, L. C. Lupo, A. Brauer

We are very grateful to the Editor Philippa Ascough, the Associate Editor Irka Hajdas, and the two anonymous reviewers for their constructive feedback that improved our manuscript. Below we detail the changes made taking into account the technical corrections specified by the Associate Editor in the "Public justification" file.

**Line 158, 193, and 235: "± 1300 BP" maybe a typo**

**Table 1: "26,500 ± 1300 BP": just curious why is this 1sig so high, or is this a typo? 130 would make sense but perhaps the sample was so very small (micrograms). This should be then explained.**

The ± 1300 BP σ value of the microbial mats from the southern shore hot spring pool is indeed high and there is no typing error. The high σ uncertainty is due to the small amount of carbon available for measurement (0.05 mgC). Nevertheless, despite the σ high value, what is relevant for this study is that the age of this sample is still the oldest.

We have added a footnote for that sample in Table 1: "*The high σ is due to the small sample size (0.05 mgC)."

**Table 1: cosmetic correction in the 1sigma significant digits (4) as the F14C and values rounding.**

Thank you for pointing out this detail. The σ values of $F^{14}C$ are now presented as 4 significant digits and rounded where appropriate.

**Line 189: insert "high F14C value of 1.1239±0.0031 (Table 1) which corresponds to the atmospheric F14C of year 1994-1996 CE (Hua et al. 2022)".**

Lines 189-193 now read as follows:

"PEI19-P-3, a woody plant of the genus *Adesmia,* was most likely dead at the time of sampling as it had no sprouts (Fig. 2a, the front plant was sampled) explaining the high $F^{14}C$ value of 1.1239 ± 0.0031 (Table 1) which corresponds to the atmospheric $F^{14}C$ of year 1994-1996 CE (Hua et al., 2022). PEI19-P-4, a Poaceae possibly *Festuca ortophylla*, dated to 2018-2019 cal CE in agreement with the sampling year (Table 1, Fig. 2b)."

**Line 274: replace "stronger" by "larger"**

We have replaced "stronger" with "larger" (now line 275).